# The intersection of the retrieval state and internal attention

**Nicole M. Long** [1] ✉

Large-scale brain states or distributed patterns of brain activity modulate downstream processing and behavior. Sustained attention and memory retrieval states impact subsequent memory, yet how these states relate to one another is unclear. I hypothesize that internal attention is a central process of the retrieval state. The alternative is that the retrieval state specifically reflects a controlled, episodic retrieval mode, engaged only when intentionally accessing events situated within a spatiotemporal context. To test my hypothesis, I developed a mnemonic state classifier independently trained to measure retrieval state evidence and applied this classifier to a spatial attention task. I find that retrieval state evidence increases during delay and response intervals when participants are maintaining spatial information. Critically, retrieval state evidence is positively related to the amount of maintained spatial location information and predicts target detection reaction times. Together, these findings support the hypothesis that internal attention is a central process of the retrieval state.

The neural mechanisms that enable access to stored information are critical to cognition, yet our understanding of these mechanisms remains limited. If you try to remember what you had for dinner last night, you might imagine yourself at the dinner table trying to picture a dish of food in front of you. You may need to engage a brain state—a temporally sustained and spatially distributed activity/connectivity pattern[1–3]—in order to access stored information about last night's dinner. However, whether the brain state that you initiate to accomplish this task is specific to memory or more generally related to attention is an open question.

The retrieval state (or mode) has been defined as a set of processes governing controlled, episodic retrieval[4]—that is, intentional access of past events situated within a spatiotemporal context, as opposed to access of general knowledge or semantic memory. The retrieval state is distinct from retrieval orientation, the specific content of the to-be-retrieved stored representation, retrieval effort, how easy or difficult it is to access the stored representation, and retrieval success, whether or not the desired representation is ultimately accessed. Engagement in the retrieval state is considered a necessary precursor for successful episodic retrieval[5,6]. Although univariate signal changes in right prefrontal cortex[5–8] have been linked to the retrieval state, recent work suggests that the retrieval state may be supported by large-scale brain networks rather than a single brain region. Specifically, distributed cortical activity patterns can reliably distinguish memory encoding and memory retrieval states[9–12] and may be driven by connectivity changes within the hippocampus[13,14].

However, the retrieval state may instead largely reflect internal attention, rather than episodic retrieval per se. Internal attention is the selection of stored representations and lies in contrast to external attention, the selection of sensory stimuli[15]. There is a broad overlap between memory and attention systems[16,17] whereby attention can be directed to memory goals and content[18] and memory, both working and long term, can guide or capture attention[19–27]. In particular, theoretical work suggests that memory retrieval may constitute a form of internal attention[15,28,29]. Computational models that include a 'spotlight of attention' can fit behavioral data from an 'episodic flanker task' in which participants perform cued recognition[29], providing foundational evidence that retrieval may constitute internal attention. The critical next step is to link, via neural measures, memory brain states to attention.

Linking memory brain states to attention is critical given the impact that both memory and attention states have on behavior. Lapses in sustained attention negatively impact subsequent memory[30–32] and engagement in a retrieval state can come at the expense of engaging an encoding state, leading to diminished subsequent memory[9]. To the

[1]Department of Psychology, University of Virginia, 22904 Charlottesville, VA, USA. ✉e-mail: niclong@virginia.edu

extent that the retrieval state extends beyond controlled, episodic memory retrieval, there are potentially wide-ranging consequences throughout cognition for engaging in this brain state. An internal attention state may facilitate decisions made based on stored information, but may similarly impair perceptual processes that depend on external attention. Furthermore, in line with extant proposals[29], demonstrating a link between internal attention and the retrieval state would suggest that memory and attention systems are subject to the same limited capacity and processing constraints.

My hypothesis is that internal attention is a central process of the retrieval state. To test this hypothesis, I developed a mnemonic state classifier, a multivariate pattern classifier trained to distinguish memory encoding vs. memory retrieval states. I have previously demonstrated that such a classifier—trained on neural signals during a mnemonic state task in which participants either encode a current stimulus or retrieve a prior stimulus while perceptual input and motor output are held constant—can be used to measure engagement in the retrieval state[9–11]. I applied this mnemonic state classifier to independent data collected in the current study in which participants perform a spatial attention task[33,34]. Participants are given a cue (left, right, or neutral) that does or does not (valid vs. invalid/neutral) predict the upcoming location of a to-be-detected probe and the stimulus onset asynchrony between cue and probe onset is variable (Fig. 1A, B). Because the classifier was trained to distinguish memory states, I refer to 'retrieval state evidence' throughout the text, although I expect that this signal is modulated by internal attention.

Here I show that (1) changing demands in the attention task modulates the retrieval state, (2) internally attended information is related to retrieval state engagement, and (3) retrieval state engagement predicts behavioral performance. These findings are consistent with the hypothesis that internal attention is a key process of the retrieval state.

## Results

### Voluntary attention impacts behavior

Consistent with prior work[35–38], I find that voluntary attention impacts behavior; namely, valid cues speed reaction times (RTs) whereas invalid cues slow RTs (Fig. 1C). I conducted a $3 \times 3$ repeated measures ANOVA (rmANOVA) with factors of cue type (valid, neutral, invalid) and stimulus onset asynchrony (SOA; 200, 400, 800 ms). Only trials in which participants responded to a target were included. There was a main effect of cue type ($F_{2,72} = 38.47$, $p < 0.001$, $\eta_p^2 = 0.52$) with the fastest responses for validly cued targets ($M = 413.5$ ms, SD = 17.42 ms), followed by neutrally cued targets ($M = 420.1$ ms, SD = 17.81 ms), and the slowest responses for invalidly cued targets ($M = 426.0$ ms, SD = 17.19 ms; valid vs. neutral, $t_{36} = -6.624$, $p < 0.001$, $d = 0.3735$, CI = [−8.595,−4.566]; neutral vs. invalid, $t_{36} = -4.247$, $p < 0.001$, $d = 0.3374$, CI = [−8.726, −3.086]; FDR corrected). There was a main effect of SOA ($F_{2,72} = 30.69$, $p < 0.001$, $\eta_p^2 = 0.46$), with the fastest responses for the 400 ms SOA (M = 412.9 ms, SD = 17.74 ms), followed by the 800 ms SOA ($M = 419.2$ ms, SD = 19.28 ms), and the slowest responses for the 200 ms SOA ($M = 427.6$ ms, SD = 16.95 ms; 200 vs. 400 ms, $t_{36} = 8.418$, $p < 0.001$, $d = 0.8521$, CI = [11.22,18.34]; 400 vs. 800 ms, $t_{36} = -3.588$, $p = 0.001$, $d = 0.3437$, CI = [−9.969,−2.769]; 200 vs. 800 ms, $t_{36} = 3.961$, $p < 0.001$, $d = 0.4634$, CI = [4.105,12.72]; FDR corrected). There was no credible evidence of an interaction between cue type and SOA ($F_{4,144} = 1.286$, $p = 0.278$, $\eta_p^2 = 0.03$). Together, these results suggest

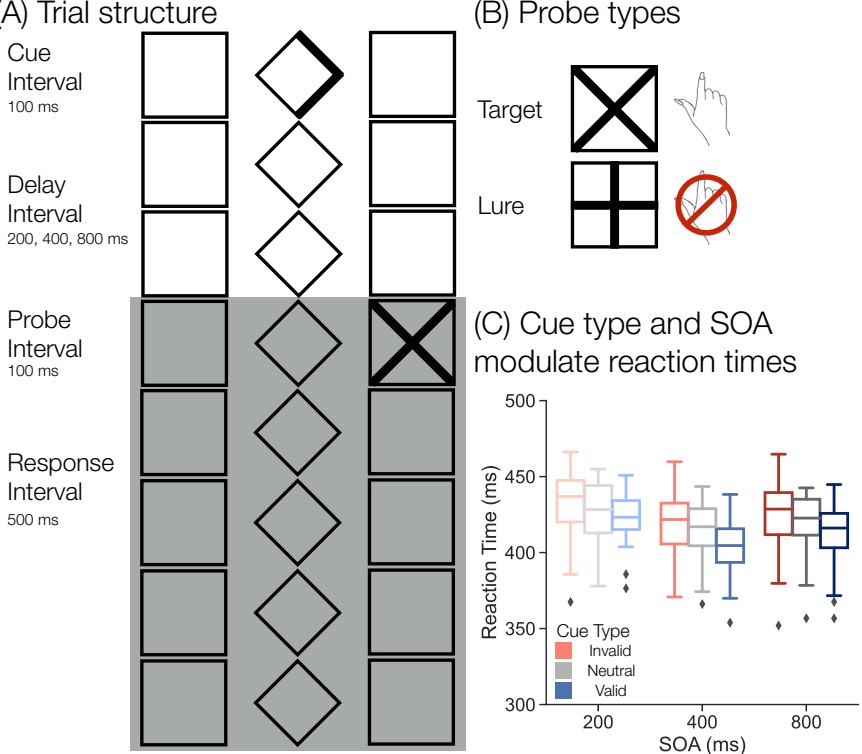

**Fig. 1 | Trial structure and behavioral results. A** To reduce exogenous attention, a central diamond and two flanking squares on either side were present throughout the session and participants were instructed to maintain central fixation throughout the study. Every trial followed this basic structure. A cue (single or double-headed arrow) is presented for 100 ms. Cue offset is followed by a variable stimulus onset asynchrony (SOA) of either 200, 400, or 800 ms (the 200 ms SOA is shown here). Following the delay interval, a probe is presented for 100 ms. Participants are given 500 ms following probe onset (shaded grey box; includes the probe interval) to provide a response. **B** If the probe was a target (an X), participants were to make a response via keyboard. If the probe was a lure (a plus), participants were to withhold their response. **C** Voluntary attention modulates behavior (n = 37 participants). Reaction times (RTs) are faster for validly (blue) cued compared to neutral (grey) trials and for neutral compared to invalidly (red) cued trials. RTs are faster for the 400 ms SOA compared to the 200 and 800 ms SOAs. Box-and-whisker plots show median (center line), upper and lower quartiles (box limits), 1.5x interquartile range (whiskers) and outliers (diamonds).

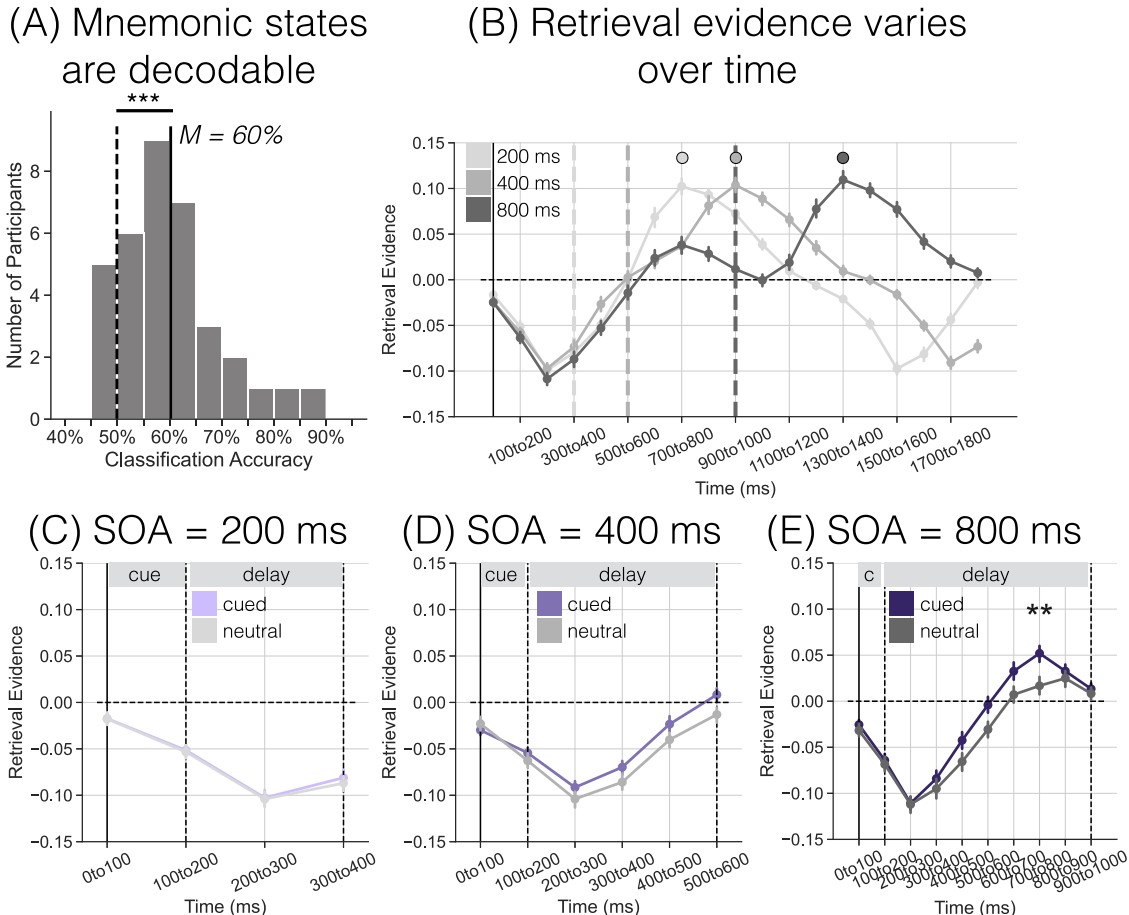

**Fig. 2 | Retrieval state evidence over time in an attention task. A** Mean classification accuracy across all participants (solid vertical line) is shown along with a histogram of classification accuracies for individual participants (gray bars) and mean classification accuracy for permuted data across all participants (dashed vertical line). Mean classification accuracy was 60%, which differed significantly from chance (two-tailed, paired *t*-test, *p* < 0.001). **B–E** Positive y-axis values indicate greater retrieval state evidence (*n* = 37 participants). The solid vertical line at time 0–100 ms indicates the onset of the cue. The vertical dashed lines indicate the onset of the probe, which varies as a function of stimulus onset asynchrony (SOA). **B** All trials are included; data have been averaged over cue type. Note that trial duration varies as a function of SOA, meaning that shorter SOA trials (200 and 400 ms) will end prior to the final time window shown. Across all SOAs, the trial initially begins with a decrease in retrieval that persists for ~500 ms, followed by an increase in retrieval that is maximal around the time point when the average response is made (indicated by the circles). **C–E** Each panel shows retrieval evidence separated by cue type (purple: cued, average of valid/invalid; grey: neutral) across the 100 ms cue and variable delay intervals. **C** The difference in retrieval state evidence between cued and neutral trials for the 200 ms SOA condition was not significant. **D** Retrieval evidence is greater for cued compared to neutral trials for the 400 ms SOA condition **E** Retrieval evidence is greater for cued compared to neutral trials for the 800 ms SOA condition. Post-hoc comparisons revealed significantly greater retrieval evidence for cued compared to neutral trials in the 700–800 ms interval (two-tailed, paired *t*-test, *p* = 0.003, FDR corrected). Error bars represent standard error of the mean. **p < 0.01; ***p < 0.001, two-tailed, paired *t*-test, FDR corrected.

that participants utilize the cue to direct voluntary attention to the cued spatial location, which facilitates target detection when the cue is valid and impairs target detection when the cue is invalid. Participants are generally faster when they have more time to prepare prior to probe onset.

**Retrieval state engagement fluctuates over time**
Given robust cross-participant classification of memory states (Fig. 2A), the first goal was to measure trial-level retrieval state evidence across time in the attention task. To the extent that internal attention is a central process of the retrieval state, there should be temporal dissociations in retrieval state evidence based on the changing attentional demands across the trial. Namely, I should find decreased retrieval evidence during the first 100 ms given that external attention will be directed to the cue. I should find increased retrieval evidence during both the delay and response intervals, given that participants must maintain perceptually absent cue or probe information during these intervals. As I am interested in retrieval state

fluctuations over time and because time intervals necessarily vary across SOAs, each SOA is evaluated separately.

I measured retrieval state evidence across nineteen 100 ms time windows, separately for each SOA and averaged across cue type (Fig. 2B). For a comprehensive view of the impact of the attention task on the retrieval state, I show all SOAs in a single figure; however, it is important to note that the 200 and 400 ms SOAs will have ended prior to the end of the time interval shown (at 1300 ms for the 200 ms SOA and at 1500 ms for the 400 ms SOA). For statistical analysis, I analyzed each SOA with custom time windows based on the trial duration. Across all SOAs, there was a main effect of time (200 ms: $F_{7,252} = 109.4$, $p < 0.001$, $\eta_p^2 = 0.75$; 400 ms: $F_{9,324} = 85.77$, $p < 0.001$, $\eta_p^2 = 0.70$; 800 ms: $F_{13,468} = 74.2$, $p < 0.001$, $\eta_p^2 = 0.67$), driven by decreased retrieval evidence early in the trial (prior to 500 ms) and increased retrieval evidence later in the trial (after 500 ms). These analyses reveal that the retrieval state is modulated by the changing demands of the attention task. Critically, if the retrieval state solely reflected controlled, episodic retrieval, there would have been no evidence for

**Table 1 | Delay interval retrieval state evidence as a function of cue type, SOA, and time, repeated measures ANOVAs**

| Effect | SOA = 200 | | | | SOA = 400 | | | | SOA = 800 | | | |
|---|---|---|---|---|---|---|---|---|---|---|---|---|
| | df | F | p | $\eta_p^2$ | df | F | p | $\eta_p^2$ | df | F | p | $\eta_p^2$ |
| Main effect of time | 1,36 | 64.82 | <0.001 | 0.64 | 3,108 | 27.34 | <0.001 | 0.43 | 7,252 | 79.5 | <0.001 | 0.69 |
| Main effect of cue | 1,36 | 0.071 | 0.792 | 0.002 | 1,36 | 4.899 | 0.033 | 0.12 | 1,36 | 9.535 | 0.004 | 0.21 |
| Interaction of time × cue | 1,36 | 0.004 | 0.952 | 0.0001 | 3,108 | 0.459 | 0.712 | 0.01 | 7,252 | 4.436 | <0.001 | 0.11 |

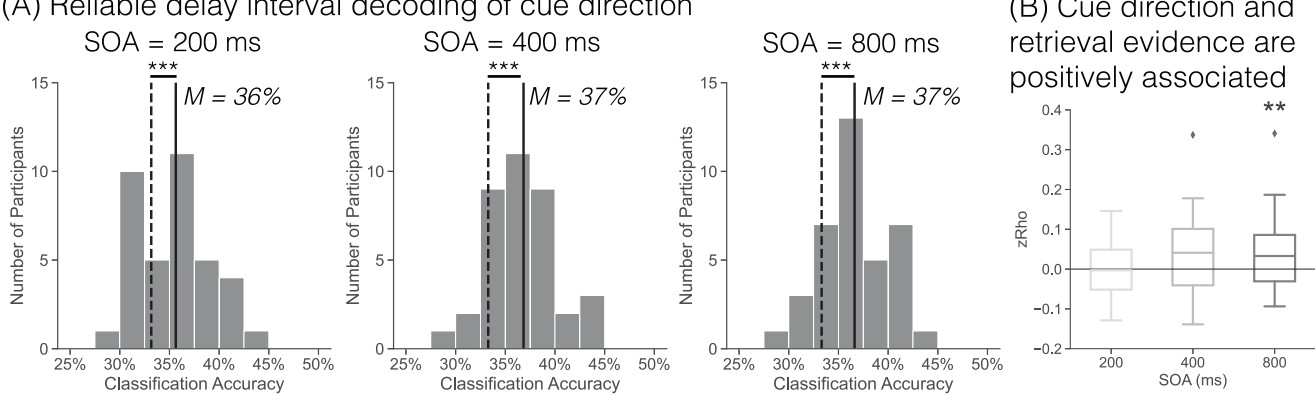

**(A) Reliable delay interval decoding of cue direction**

**(B) Cue direction and retrieval evidence are positively associated**

**Fig. 3 | Delay interval cue direction and memory state.** Spectral power was averaged over the respective delay intervals (100–300, 100–500, 100–900 ms) for the current analyses. **A** Each histogram shows the classification accuracy for within participant leave-one-run-out cross validated classification of the cue direction (left, right, neutral) during the delay interval for each SOA condition. Cue direction decoding during the delay interval is significantly above chance (two-tailed, paired *t*-test, *p*'s < 0.001, as determined by permutation procedure). **B** I performed trial level Pearson correlations between cue direction evidence (left or right with neutral evidence as a baseline, see Methods) and retrieval state evidence (*n* = 37 participants). There is a significant correlation between cue direction evidence and retrieval state evidence for the 800 ms SOA condition (two-tailed, paired *t*-test, *p* = 0.005, FDR corrected). Box-and-whisker plots show median (center line), upper and lower quartiles (box limits), 1.5x interquartile range (whiskers) and outliers (diamonds). \*\**p* < 0.01; \*\*\**p* < 0.001, two-tailed, paired *t*-test, FDR corrected.

retrieval state modulation given that there are no episodic memory demands in the attention task.

**Voluntary attention modulates delay interval retrieval evidence**
Maintaining spatial location across a delay facilitates target detection[39] and spatial attention can be directed to internal representations in working memory[40]. To the extent that internal attention is a central process of the retrieval state, I expect to find that delay interval retrieval state evidence is greater for cued (valid and invalid) compared to neutral trials.

For each SOA, I averaged retrieval evidence across valid and invalid trials, as prior to the probe, valid and invalid trials are equivalent (Fig. 2C–E; Table 1). For all three SOAs, there was a main effect of time, indicating that retrieval state engagement fluctuates over the delay interval, consistent with the full trial analysis above. For both the 400 and 800 ms SOAs, there was a significant main effect of cue type, with greater retrieval state evidence for cued compared to neutral trials (400 ms: cued, $M = -0.0598$, $SD = 0.0325$, neutral, $M = -0.0732$, $SD = 0.0373$, $t_{36} = 2.213$, $p = 0.033$, $d = 0.3833$, CI = [0.0011,0.0257]; 800 ms: cued, $M = -0.0235$, $SD = 0.0351$, neutral, $M = -0.0404$, $SD = 0.0391$, $t_{36} = 3.088$, $p = 0.004$, $d = 0.4532$, CI = [0.0058,0.0279]). For the 800 ms SOA, there was a significant interaction between cue type and time, driven by greater retrieval evidence for cued compared to neutral trials from 500–800 ms, with the 700–800 ms time window surviving FDR correction (500–600 ms: $t_{36} = 2.427$, $p = 0.020$, $d = 0.5127$, CI = [0.0044,0.0488]; 600–700 ms: $t_{36} = 2.317$, $p = 0.026$, $d = 0.4532$, CI = [0.0032,0.0477]; 700–800 ms: $t_{36} = 3.224$, $p = 0.003$, $d = 0.6330$, CI = [0.0131,0.0574]). To directly compare delay interval retrieval evidence across SOAs, I averaged retrieval evidence over the delay interval and performed a 2 (cued/neutral) × 3 (SOA) rmANOVA. There was a main effect of cue ($F_{1,36} = 6.524$, $p = 0.015$, $\eta_p^2 = 0.15$), driven by greater retrieval evidence for cued compared to neutral trials. There was a main effect of SOA

($F_{2,72} = 30.97$, $p < 0.001$, $\eta_p^2 = 0.46$), with increasing retrieval evidence over longer delays. There was no credible evidence for a cue by SOA interaction ($F_{2,72} = 2.599$, $p = 0.081$, $\eta_p^2 = 0.07$). Together, these findings demonstrate that voluntary attention is linked with retrieval state engagement.

Having shown that retrieval evidence is modulated by voluntary attention, I next sought to directly test the relationship between the retrieval state and the information to which attention is directed. During the delay interval, internal attention should be directed to the cue location (left/right on cued trials). If internal attention is a central process of the retrieval state, then retrieval state evidence should scale with cue location evidence.

I first established that cue direction (left, right, neutral) can be decoded during the delay interval (Fig. 3A). For each SOA, I averaged spectral power across the respective delay interval (100–300, 100–500, or 100–900 ms). I performed within participant leave-one-run-out cross-validated classification. For each SOA, I found significantly above chance (as determined through permutation procedure, see Methods) classification accuracy (200 ms: $M = 35.64\%$, SD = 3.55%, $t_{36} = 4.088$, $p < 0.001$, $d = 0.975$, CI = [0.0124,0.037]; 400 ms: $M = 36.84\%$, SD = 3.73%, $t_{36} = 5.838$, $p < 0.001$, $d = 1.344$, CI = [0.0232,0.0479]; 800 ms: $M = 36.62\%$, SD = 3.17%, $t_{36} = 6.294$, $p < 0.001$, $d = 1.474$, CI = [0.0225,0.0438]). Thus, it is possible to reliably decode to which cue direction (left, right, or neither) participants are attending during the delay interval.

I next tested the relationship between cue direction and retrieval evidence (Fig. 3B). For each participant, I extracted left/right cue direction evidence (using neutral evidence as a baseline, see Methods) and retrieval evidence during the delay interval, separately for each SOA. For each participant, I performed a Pearson correlation across trials, separately for left and right cues, and averaged Fisher *Z* transformed rho values across cue direction. There was a significant positive correlation for the 800 ms SOA (zRho = 0.0408, $t_{36} = 2.969$, $p = 0.005$,

$d = 0.4948$, CI = [0.0129,0.0687]). Although numerically positive, there was not a significant correlation for the 200 or 400 ms SOA (200 ms: $zRho = 0.0019$, $t_{36} = 0.1533$, $p = 0.879$, $d = 0.3212$, CI = [−0.0227, 0.0264]; 400 ms: $zRho = 0.0319$, $t_{36} = 1.9271$, $p = 0.062$, $d = 0.0256$, CI = [−0.0017,0.0655]). A 1 × 3 (SOA) rmANOVA with zRho as the dependent variable did not reach significance ($F_{2,72} = 2.95$, $p = 0.059$, $\eta_p^2 = 0.08$); average zRho across SOAs was significantly greater than zero ($zRho = 0.0249$, $t_{36} = 2.378$, $p = 0.023$, $d = 0.3964$, CI = [0.0037, 0.046]). These results indicate that cue information and retrieval evidence are generally positively associated across the delay interval.

## Retrieval evidence fluctuates during the response interval

To the extent that internal attention is a central process of the retrieval state, retrieval state evidence should be modulated during the response interval. The probe (a plus or a cross) is shown for 100 ms following the delay interval. The participant's task is to make or with-hold a response within 500 ms of probe onset, depending on which probe is presented. Importantly, since the probe is presented for only 100 ms, the participant must hold in mind information about the probe in order to make a decision. Such a condition should require internally directed attention to the perceptually absent probe. I therefore expect that across all trials—regardless of cue type, SOA, and probe type—retrieval state evidence will increase leading up to the time in which a response is made or withheld. I further anticipate that cue type and SOA will modulate response interval retrieval state evidence. Specifically, valid cues should enable participants to attend to the correct location prior to probe onset and reduce or eliminate demands to re-orient attention to a different location. These reduced external attention demands may facilitate engagement of the retrieval state. For shorter SOAs, participants are likely still processing the cue and thus may exhibit less retrieval evidence for those SOAs.

I assessed probe-locked retrieval state evidence during the response interval (Fig. 4A) via a 3 × 3 × 5 rmANOVA with factors of cue type (valid, neutral, invalid), SOA (200, 400, 800 ms), and time window (0–500 ms in five 100 ms windows). I report the full results in Table 2 and highlight the follow-up tests here. The three way interaction between cue type, SOA and time window was not significant and Bayes Factor analysis revealed that a model without the three-way interaction term is preferred to a model with the three-way interaction term by a factor of 1.267 × 10¹⁹. All other main effects and interactions were significant.

I expected retrieval evidence to increase leading up to the response as participants direct internal attention to the maintained probe. There was a main effect of time whereby retrieval evidence was maximal around the time of the average response (400–500 ms). I performed post-hoc one sample $t$-tests at each time window and found a significant decrease in retrieval evidence during the first two time windows (0–100 ms: $t_{36} = −5.441$, $p < 0.001$, $d = 0.8945$, CI = [−0.0321,−0.0147]; 100–200 ms: $t_{36} = −2.338$, $p = 0.025$, $d = 0.3844$, CI = [−0.0241,−0.0017]; FDR corrected) and a significant increase in retrieval evidence during the latter three time windows (200–300 ms: $t_{36} = 2.260$, $p = 0.030$, $d = 0.3715$, CI = [0.0017,0.0312]; 300–400 ms: $t_{36} = 7.537$, $p < 0.001$, $d = 1.239$, CI = [0.0545,0.0946]; 400–500 ms: $t_{36} = 12.82$, $p < 0.001$, $d = 2.108$, CI = [0.0882,0.1213]; FDR corrected). Thus, over time, engagement in the retrieval state increases, possibly reflecting increased internal attention directed to information about the probe.

By preemptively directing attention to the correct probe location, valid—relative to invalid and neutral—cues may speed entrance into a retrieval state, leading to an increase in retrieval evidence at earlier time intervals. There was a main effect of cue type and an interaction between cue type and time whereby retrieval evidence was greater for valid compared to invalid and neutral trials within the first 200 ms of the response interval (Fig. 4A, left panel). Post-hoc 1 × 3 (cue) rmANOVAs at each time window revealed a significant effect of cue from 0–200 ms (0–100 ms: $F_{2,72} = 5.171$, $p = 0.008$, $\eta_p^2 = 0.1256$; 100–200 ms:

$F_{2,72} = 7.007$, $p = 0.002$, $\eta_p^2 = 0.1629$; FDR corrected). Follow-up paired $t$-tests revealed significantly greater retrieval evidence for valid compared to invalid trials in the 100–200 ms window ($t_{36} = 3.221$, $p = 0.003$, $d = 0.4386$, CI = [0.0065,0.0288] FDR corrected) and for valid compared to neutral trials from 0–200 ms (0–100 ms: $t_{36} = 3.824$, $p < 0.001$, $d = 0.5433$, CI = [0.0072,0.0234]; 100–200 ms: $t_{36} = 3.693$, $p < 0.001$, $d = 0.4891$, CI = [0.0071,0.0245]; FDR corrected). There was no credible evidence for differences between neutral and invalid trials ($t$'s < 1.2, $p$'s > 0.26). The main effect of cue did not reach significance for the later time windows (200–300 ms: $F_{2,72} = 2.980$, $p = 0.057$, $\eta_p^2 = 0.0764$; 300–400 ms: $F_{2,72} = 0.6425$, $p = 0.529$, $\eta_p^2 = 0.0175$; 400–500 ms: $F_{2,72} = 0.6953$, $p = 0.502$, $\eta_p^2 = 0.0189$). These results show that valid cues facilitate entrance into a retrieval state. It may be that less external attention is needed to process the probe or that no reorientation of external attention (to a different location) is needed on valid trials. Reduced demand to either switch into an external state and/or to reorient external attention to a different spatial location may result in the greater degree of retrieval state evidence on validly cued trials.

To the extent that participants are still directing external attention to the just-presented cue, entrance into a retrieval state should be delayed specifically for the 200 ms SOA. There was a main effect of SOA and an interaction between SOA and time whereby retrieval evidence was decreased for the 200 compared to 400 and 800 ms SOAs (Fig. 4A, middle panel). I performed post-hoc 1 × 3 (SOA) rmANOVAs at each time window and find a significant effect of SOA for all time windows excluding the final 400–500 ms window (0–100 ms: $F_{2,72} = 65.25$, $p < 0.001$, $\eta_p^2 = 0.6445$; 100–200 ms: $F_{2,72} = 45.76$, $p < 0.001$, $\eta_p^2 = 0.5597$; 200–300 ms: $F_{2,72} = 13.86$, $p < 0.001$, $\eta_p^2 = 0.278$; 300–400 ms: $F_{2,72} = 3.460$, $p = 0.037$, $\eta_p^2 = 0.0877$; 400–500 ms: $F_{2,72} = 2.414$, $p = 0.097$, $\eta_p^2 = 0.0628$; FDR corrected). Follow-up paired $t$-tests revealed significantly decreased retrieval evidence for the 200 compared to 400 ms SOA in all four time windows (0–100 ms: $t_{36} = −8.781$, $p < 0.001$, $d = 1.920$, CI = [−0.1037,−0.0648]; 100–200 ms: $t_{36} = −9.491$, $p < 0.001$, $d = 1.693$, CI = [−0.0884,−0.0572]; 200–300 ms: $t_{36} = −5.931$, $p < 0.001$, $d = 0.796$, CI = [−0.0511,−0.0251]; 300–400 ms: $t_{36} = −2.549$, $p = 0.015$, $d = 0.2589$, CI = [−0.0296,−0.0034]; FDR corrected) and for the 200 compared to 800 ms SOA from 0–300 ms (0–100 ms: $t_{36} = −9.831$, $p < 0.001$, $d = 2.355$, CI = [−0.1143,−0.0752]; 100–200 ms: $t_{36} = −6.483$, $p < 0.001$, $d = 1.240$, CI = [−0.0686,−0.0359]; 200–300 ms: $t_{36} = −2.669$, $p = 0.011$, $d = 0.4448$, CI = [−0.0392,−0.0053]; 300–400 ms: $t_{36} = −1.864$, $p = 0.071$, $d = 0.2229$, CI = [−0.0302,0.0013]; FDR corrected). Retrieval evidence was significantly greater for the 400 compared to 800 ms SOA from 100–300 ms (100–200 ms: $t_{36} = 2.63$, $p = 0.013$, $d = 0.4539$, CI = [0.0047,0.0363]; 200–300 ms: $t_{36} = 2.292$, $p = 0.028$, $d = 0.2921$, CI = [0.0018,0.0299]; FDR corrected), but not for the remaining time windows ($t$'s < 1.4, $p$'s > 0.15). These findings support the interpretation that external attention is still focused on the cue during the 200 ms SOA which diminishes overall engagement of the retrieval state on these trials.

Finally, there was an interaction between cue type and SOA driven by greater retrieval evidence on valid trials during the shortest SOA. I performed post-hoc 1 × 3 (cue) rmANOVAs for each SOA and find a significant effect of cue for the 200 ms SOA (200 ms: $F_{2,72} = 7.943$, $p < 0.001$, $\eta_p^2 = 0.1808$; FDR corrected). Retrieval evidence was significantly greater for valid compared to both invalid ($t_{36} = 4.117$, $p < 0.001$, $d = 0.5897$, CI = [0.0133,0.039], FDR corrected) and neutral ($t_{36} = 2.617$, $p = 0.013$, $d = 0.3936$, CI = [0.003,0.0258], FDR corrected) trials, but did not significantly differ between neutral and invalid trials ($t_{36} = 1.518$, $p = 0.138$, $d = 0.2673$, CI = [−0.0271,0.0039]). There was no significant effect of cue for the other SOAs (400 ms: $F_{2,72} = 1.564$, $p = 0.216$, $\eta_p^2 = 0.0416$; 800 ms: $F_{2,72} = 0.3399$, $p = 0.713$, $\eta_p^2 = 0.0094$). These results are consistent with the prior analyses showing that valid cues induce stronger retrieval state engagement relative to neutral and invalid trials.

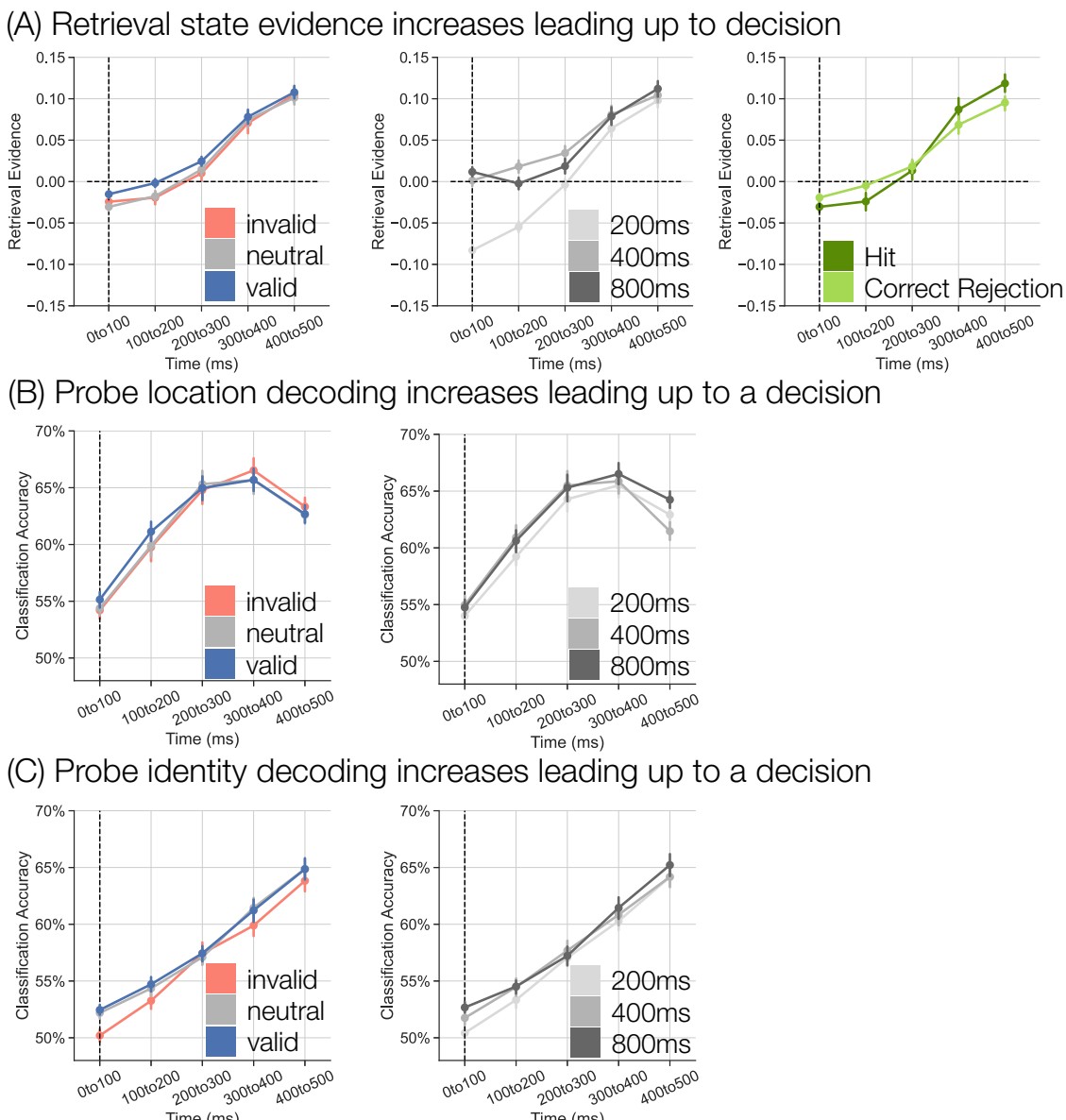

**Fig. 4 | Response interval retrieval state and probe classification accuracy.**
**A** Each panel shows probe-locked retrieval state evidence; positive values indicate greater retrieval state evidence ($n = 37$ participants). The dashed vertical line at time 0–100 ms indicates the onset of the probe (cross target or plus lure). The left panel shows retrieval state evidence separated by cue type (invalid, red; neutral, grey; valid, blue). The middle panel shows retrieval state evidence separated by SOA (200, 400, 800 ms). The right panel shows retrieval state evidence separately for hits (cross targets to which participants responded; dark green) and correct rejections (plus lures to which participants withheld a response; light green). **B** Classification accuracy for within participant leave-one-run-out cross validated classification of the probe location ($n = 37$ participants). **C** Classification accuracy for within participant leave-one-run-out cross validated classification of probe identity ($n = 37$ participants). Error bars represent standard error of the mean.

**Table 2 | Response interval retrieval state evidence and probe decoding accuracy as a function of cue type, SOA, and time, repeated measures ANOVAs**

| Effect | Retrieval Evidence | | | | Probe Location | | | Probe Identity | | |
|---|---|---|---|---|---|---|---|---|---|---|
| | df | F | p | $\eta_p^2$ | F | p | $\eta_p^2$ | F | p | $\eta_p^2$ |
| Main effect of time | 4,144 | 113.2 | <0.001 | 0.76 | 81.6 | <0.001 | 0.69 | 138.1 | <0.001 | 0.79 |
| Main effect of cue | 2,72 | 3.208 | 0.046 | 0.08 | 0.313 | 0.732 | 0.0086 | 9.269 | <0.001 | 0.20 |
| Main effect of SOA | 2,72 | 36.68 | <0.001 | 0.50 | 3.896 | 0.025 | 0.10 | 5.067 | 0.009 | 0.12 |
| Interaction of time × cue | 8,288 | 2.228 | 0.026 | 0.06 | 1.378 | 0.206 | 0.04 | 1.153 | 0.328 | 0.03 |
| Interaction of time × SOA | 8,288 | 26.74 | <0.001 | 0.43 | 2.847 | 0.005 | 0.07 | 1.215 | 0.290 | 0.03 |
| Interaction of cue × SOA | 4,144 | 2.687 | 0.034 | 0.07 | 1.09 | 0.364 | 0.03 | 1.559 | 0.189 | 0.04 |
| Interaction of cue × SOA × time | 16,576 | 0.625 | 0.865 | 0.02 | 0.859 | 0.617 | 0.02 | 0.739 | 0.755 | 0.02 |

## (A) Probe location and retrieval evidence are positively correlated

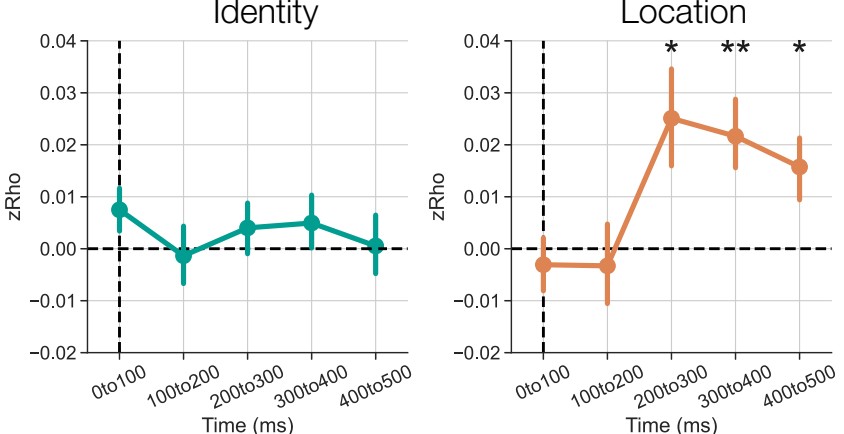

## (B) Retrieval and probe identity evidence predict reaction times

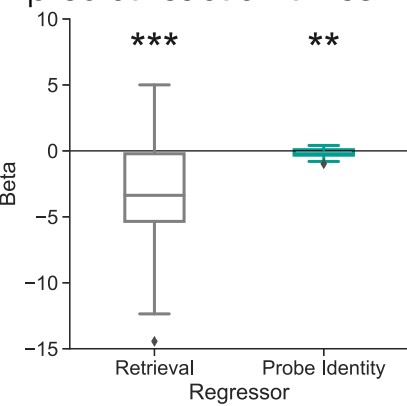

**Fig. 5 | Relationship between probe information, retrieval evidence, and behavior. A** I performed trial level Pearson correlations between probe identity (cross, plus) evidence (left panel) or probe location (left, right) evidence (right panel) and retrieval state evidence across the response interval ($n = 37$ participants). There is a significant positive correlation between probe location and retrieval evidence during the 200–500 ms of the response interval (two-tailed, paired $t$-tests, 200–300 ms: $p = 0.042$, 300–400 ms: $p = 0.003$, 400–500 ms: $p = 0.016$, FDR corrected). Error bars represent standard error of the mean. **B** I performed multiple linear regression in which I used retrieval evidence and probe identity evidence during the 300–400 ms time window to predict reaction times (RTs; $n = 37$ participants). Only trials with RTs > 400 ms are included. There were significant negative betas for both regressors (two-tailed, paired $t$-tests, retrieval evidence: $p < 0.001$, probe identity evidence: $p = 0.001$, FDR corrected) meaning that more retrieval evidence and more probe identity evidence predict faster RTs. Box-and-whisker plots show median (center line), upper and lower quartiles (box limits), 1.5x interquartile range (whiskers) and outliers (diamonds). *$p < 0.05$; **$p < 0.01$; ***$p < 0.001$; two-tailed paired $t$-tests, FDR corrected.

Taken together, the assessment of the retrieval state during the response interval reveals modulations consistent with the interpretation that the retrieval state reflects internal attention. Namely, valid cues and longer SOAs promote retrieval state engagement, potentially by reducing external attention demands either to the probe or the cue, respectively.

My interpretation is that the increase in retrieval state evidence over time reflects internal attention to the probe; however, it is possible that the retrieval evidence increase is driven by preparation for a motor movement. I can directly test this alternative by comparing retrieval state evidence as a function of response condition. I assessed retrieval evidence for hits (cross trials to which participants responded) and correct rejections (CRs; plus trials to which participants withheld a response; Fig. 4A, right panel). I conducted a 2 × 5 rmANOVA with condition (hit, CR) and time window as factors. There was no credible evidence for a main effect of condition ($F_{1,36} = 0.018$, $p = 0.894$, $\eta_p^2 = 0.0004$). There was a main effect of time ($F_{4,144} = 108.9$, $p < 0.001$, $\eta_p^2 = 0.75$). There was a significant interaction between condition and time ($F_{4,144} = 12.65$, $p < 0.001$, $\eta_p^2 = 0.26$). Numerically, relative to CRs, hits are characterized by decreased retrieval evidence in the first 200 ms and increased retrieval evidence in the last two 200 ms of the response interval; however there are no significant differences in retrieval evidence that survive multiple comparisons correction ($t$'s < 2.23, $p$'s > 0.03). That I find an increase in retrieval evidence even when no response is made provides evidence against the interpretation that retrieval evidence is solely driven by motor responses. Instead, retrieval fluctuates dynamically over time depending on the decision made and increases when a decision is made based on internal information, regardless of whether an actual response is made.

### Probe information fluctuates during the response interval
To the extent that the retrieval state reflects internal attention, information maintained during the response interval—probe location or identity—should be related to retrieval evidence. This is analogous to

the link that was demonstrated between delay interval retrieval and cue direction evidence.

I first established that probe location (left, right) and probe identity (cross, plus) can be reliably decoded during the response interval. I performed within participant leave-one-run-out cross-validated classification on spectral signals averaged across the response interval (0–500 ms relative to probe onset). I find significantly above chance classification accuracy of both probe location ($M = 65.44\%$, SD = 6.14%, $t_{36} = 15.06$, $p < 0.001$, $d = 3.555$, CI = [0.1336,0.1753]) and probe identity ($M = 60.33\%$, SD = 5.14%, $t_{36} = 12.09$, $p < 0.001$, $d = 2.847$, CI = [0.0862,0.1209]).

As was the case with retrieval evidence, I expect probe information to vary across the response interval and as a function of cue type and SOA. Specifically, I expect probe information to increase leading up to the response and that decoding accuracy will be higher when participants can preallocate internal attention—that is, during valid trials and longer SOAs. I assessed the impact of cue type, SOA, and time on classification accuracy from the probe location and identity classifiers (Fig. 4B, C; Table 2). I performed within participant leave-one-run-out cross-validated classification across each of the five 100 ms time windows. I then back-sorted trials by cue type and SOA to test for dissociations in classification accuracy. I conducted 3 × 3 × 5 rmANOVAs with cue type, SOA, and time window as factors.

Probe information was modulated by time and selectively by cue type and SOA, depending on the type of information. Classification accuracy was consistently modulated by time whereby both identity and location accuracy was higher during later time windows. Location accuracy was modulated by SOA whereby location accuracy was greater for the 800 compared to 200 ms SOA ($t_{36} = 3.302$, $p = 0.002$, $d = 0.2468$, CI = [0.0042,0.0177], FDR corrected) and did not significantly differ between the other two SOAs (200 vs. 400 ms: $t_{36} = -1.257$, $p = 0.217$, $d = 0.1123$, CI = [−0.0145,0.0034]; 400 vs. 800 ms: $t_{36} = -1.3642$, $p = 0.181$, $d = 0.1094$, CI = [−0.0134,0.0026]). Identity accuracy was modulated by cue type whereby identity accuracy was greater for valid relative to invalid trials ($t_{36} = 3.905$, $p < 0.001$,

$d = 0.3184$, CI = [0.0059,0.0185]; FDR corrected) and for neutral relative to invalid trials ($t_{36} = 3.2243$, $p = 0.003$, $d = 0.2939$, CI = [0.004,0.0175]; FDR corrected), but did not significantly differ for valid vs. neutral trials ($t_{36} = 0.515$, $p = 0.610$, $d = 0.0388$, CI = [−0.0042,0.0071]). Identity accuracy was also modulated by SOA whereby identity accuracy was greater for the 800 relative to 200 ms SOA ($t_{36} = 3.030$, $p = 0.005$, $d = 0.3167$, CI = [0.0038,0.0194], FDR corrected), but did not significantly differ between the other SOAs (200 vs. 400 ms: $t_{36} = −2.011$, $p = 0.052$, $d = 0.1922$, CI = [−0.0147,0.0001]; 400 vs. 800 ms: $t_{36} = −1.198$, $p = 0.239$, $d = 0.1102$, CI = [−0.0116,0.003]). Together, these results mirror the retrieval state findings, namely that with accurate cue information (valid trials) and with more time to prepare (longer SOAs), probe information is of higher fidelity−more decodable−during the response interval.

### Probe maintenance engages retrieval and retrieval facilitates responses

Having shown that both retrieval evidence and probe information fluctuate across the response interval, I next tested the relationship between probe information and retrieval state evidence. I performed Pearson correlations between either probe identity evidence (Fig. 5A, left panel) or probe location evidence (Fig. 5A, right panel) and retrieval evidence at each time window. I performed the correlations across all trials irrespective of cue type or SOA as I expect a positive association between probe information and retrieval state across all conditions. There were no significant correlations between identity and retrieval evidence ($t$'s < 1.9, $p$'s > 0.06). There was a significant positive correlation between location and retrieval evidence from 200−500 ms (200−300 ms: $t_{36} = 2.578$, $p = 0.014$, $d = 0.4297$, CI = [0.0054,0.0448]; 300−400 ms: $t_{36} = 3.191$, $p = 0.003$, $d = 0.5318$, CI = [0.0079,0.0354]; 400−500 ms: $t_{36} = 2.536$, $p = 0.016$, $d = 0.4227$, CI = [0.0032,0.0283]; FDR corrected), but not during the first two time windows ($t$'s < 0.60, $p$'s > 0.55). These results show that as retrieval evidence increases, so too does information about the probe's location.

My final goal was to link retrieval state evidence and behavior. It is possible that retrieval evidence builds up over time when no external information is present (e.g. during the delay and response intervals), but is not directly related to behavior. If the retrieval state reflects attention directed to different sources of internal information (cue or probe), as shown in the analyses above, it should predict RTs. I expect probe identity information to also predict RTs, given that this information is needed to make a decision. I focused specifically on evidence from 300−400 ms as this time window immediately precedes the majority of responses (64% of responses were made during the 400−500 ms window).

I conducted a multiple linear regression analysis with retrieval evidence and identity evidence during the 300−400 ms time window as regressors (Fig. 5B). I omitted location evidence given its correlation with retrieval evidence. I only included target trials with RTs > 400 ms. I modeled all trials together irrespective of cue type and SOA to maximize the number of trials available for the regression estimates and because I expect that more evidence should predict faster RTs regardless of cue type and SOA. I find significant negative betas for both regressors. Increases in retrieval evidence predicts faster RTs ($M = −3.205$, SD = 4.233, $t_{36} = 4.543$, $p < 0.001$, $d = 0.7572$, CI = [1.774,4.636]), as do increases in identity evidence ($M = −0.1875$, SD = 0.3232, $t_{36} = 3.481$, $p = 0.001$, $d = 0.5801$, CI = [0.0782,0.2967]). Together these findings demonstrate that both retrieval evidence and information about probe identity facilitate target responses.

## Discussion

The aim of the present study was to test the hypothesis that internal attention is a central process of the retrieval state. I used multivariate pattern analysis across two independent studies to measure retrieval state engagement in a spatial attention task with no episodic memory demands. I find retrieval state fluctuations in response to external and internal attention demands and specifically that retrieval state evidence increases during periods of time when internal attention should be employed and no external information is present. Critically, I find that increases in retrieval state evidence relate to the information being attended and predict faster reaction times (RTs), linking the retrieval state to behavior. Together, these findings demonstrate that internal attention constitutes a central process of the retrieval state, which has implications for the role of this brain state across many cognitive contexts.

Retrieval state evidence fluctuates throughout the trial in the attention task. At the broadest level, if the retrieval state solely reflected controlled, episodic retrieval[4], then there should have been no retrieval state modulation in the current study. Given that retrieval state evidence decreased in response to the cue and increased during the delay and response intervals, this provides support for the hypothesis that internal attention is a central process of the retrieval state. Internal attention is the selection of the stored contents of the mind, including working and long term memory representations[15]. My interpretation is that participants direct their mind's eye inwards at multiple points in a given trial in the attention task and that retrieval evidence tracks the extent to which participants have selected internal representations. This interpretation is in line with recent theoretical models in which an internal attentional spotlight supports memory retrieval by focusing attention to stored episodic representations[29].

Prior work has proposed that internal attention is responsible for selecting relevant features of a memory after an episodic retrieval mode has been established[41]. In this view, internally directed attention is more in line with retrieval orientation than the retrieval mode. However, if episodic retrieval had to precede the act of turning the mind's eye inward, I would not have observed any retrieval state fluctuations in the present study, as there were no episodic retrieval demands. My interpretation is that attention must be directed internally before one can retrieve an episodic memory and that the episodic component constitutes the orientation rather than the state. It is more parsimonious to posit a single internal attention state that is needed for both episodic and semantic retrieval, since both rely on stored representations. That episodic and semantic retrieval recruit highly overlapping neural substrates−in particular, the default mode network[42,43], which in turn has been directly linked to internal attention[44]−provides evidence in support of this account. Internal attention may still be necessary for successful episodic retrieval−the original proposed function of the retrieval mode−but internal attention will also be necessary for selecting any form of internal information, regardless of the specific content.

Beyond fluctuating over time, I have directly linked the retrieval state to the internal information that is selected. When participants maintain cue direction during the delay, retrieval evidence increases and is positively related to information about the cue direction. Similarly, when participants maintain probe information during the response interval, retrieval evidence increases leading up to the response and is positively related to probe location information. Working memory maintenance may serve as the mechanism that links the retrieval state and internal attention. There is an intimate connection between selective attention, working memory, and long term memory[16]. In particular, selective attention and working memory processes recruit shared neural substrates[17,39,45–47]. Furthermore, univariate working memory maintenance signals are engaged during long term memory retrieval[48] and multivariate perceptual representations are reinstated both during working memory maintenance and long term memory retrieval[49]. Taken together, these findings support the interpretation that working memory may be recruited for maintenance of both the retrieved stimuli in the mnemonic state task and the cue/probe spatial information in the attention task. A prediction from this

interpretation is that retrieval state engagement should be observed in any task which includes/relies on working memory maintenance.

I find that as spatial information fidelity increases (whether cue direction or probe location), retrieval evidence correspondingly increases. Spatial attention and spatial working memory are clearly linked to spectral power in the alpha band[50–52]. Specifically, posterior alpha power is decreased over the hemisphere contralateral to the visual hemifield to which attention (internal or external) is directed[53]. The mnemonic state classifier includes posterior alpha power among its features, along with other electrodes and frequencies. Given that retrieval evidence correlates to spatial information, one may ask whether the retrieval state is simply indexing spatial working memory. To the extent that spatial information is what is being maintained and/or attended, retrieval state evidence should reflect these demands. However, it is unlikely that the retrieval state exclusively or solely reflects spatial working memory. Note that the mnemonic state classifier I developed was trained on centrally presented object stimuli with no spatial attention demands. Although location information may be automatically encoded or retrieved[54], this spatial information would not differentiate encode and retrieve trials and therefore is unlikely to drive classifier performance. Furthermore, the classifier had no a priori information about left vs. right spatial locations and thus cannot purely reflect such information. Instead, my interpretation is that the retrieval state reflects any form of internally directed attention, whether to spatial, episodic, semantic, or other perceptual information that is not currently present in the external environment. However, future work is necessary to directly test the extent to which the retrieval state is driven by maintenance of spatial vs. non-spatial information.

Increases in retrieval state evidence predicts faster RTs. This extends prior work showing that memory states impact memory behavior[9] and indicates that memory states also influence attention-based perceptual decisions. Pre-allocating voluntary attention can facilitate behavioral responses[55]. Here I show that retrieval evidence increases leading up to a decision and facilitates responses. Via internal attention processing, the retrieval state may track and predict numerous behaviors that depend on accessing stored information.

The current findings suggest an intimate connection in the neural mechanisms between memory and attention systems, particularly in regards to large-scale brain activity patterns or states. A critical next step will be to perform direct cross-task classification on a set of a retrieval and internal attention tasks in order to determine the extent to which the two tasks overlap. Robust cross-task classification would provide evidence that the retrieval state and internal attention are one and the same, whereas the current findings leave open the possibility that specific processes beyond internal attention are unique to the retrieval state. Conceptually, reframing the retrieval state as an internal attention state would reduce the parameter space with which these systems can be understood[29] and would suggest that both systems share processing constraints and are subject to the same capacity limitations. In practical terms, the approach and data utilized in this manuscript can be applied to any dataset with the same features (63 electrodes and 46 frequencies) in order to estimate retrieval state engagement. By applying the mnemonic state classifier used here, it will be possible to measure how retrieval impacts not just memory and attention processes, but also decision making—e.g. how online evidence accumulation impacts decisions[56], how stored information is used to inform value-based decisions[57], and how persistence in a retrieval state may impair perception of external information.

An important direction for future work will be to define the boundary conditions between external attention and the memory encoding state. Because of the structure of the mnemonic state classifier, a decrease in retrieval evidence is synonymous with an increase in encoding evidence. Although I expect that at least some elements of the encoding state reflect external attention, I cannot address this question in the present manuscript as I intentionally did not include a memory test so as to eliminate episodic memory demands. Given the persistence of decreased retrieval state evidence for several hundred milliseconds after cue presentation in the current study, I anticipate that such (relatively) slow state changes may account for phenomena such as the attentional blink[58]. The inability to redirect attention to a second target presented in close succession to a previous target may be connected with the need to shift between large-scale brain states. Furthermore, the present work raises questions regarding the temporal dynamics of memory brain states, including how individuals transition into and out of these brain states and the consequences for behavior of switching vs. staying in a particular brain state.

In summary, I demonstrate that the retrieval state, a neural brain state supported by distributed cortical activity patterns, is largely driven by internal attention. Retrieval evidence is modulated by voluntary attention and facilitates behavioral responses in an attention-based target detection task. Engagement of internal attention as part of the retrieval state means that this brain state is likely engaged and influencing behavior throughout cognition, opening new avenues for critical work to investigate how brain states impact cognition.

## Methods

This research complies with all relevant ethical regulations and was approved by the University of Virginia Institutional Review Board for Social and Behavioral Research.

### Participants

Forty (31 female; ages 18–28, mean age = 20.25 years), native English speakers from the University of Virginia community participated. Gender was determined based on self-report and no gender based analyses were conducted as there were no expectations that any effects would covary with gender. All participants had normal or corrected-to-normal vision. Informed consent was obtained in accordance with the University of Virginia Institutional Review Board for Social and Behavioral Research and participants were compensated for their participation. No statistical method was used to predetermine sample size; sample size was based on prior work[9]. Three participants were excluded from the analyses. One participant was excluded due to data loss. Two participants were excluded due to poor signal quality (impedances were above the threshold of 50 kΩ). Data are reported from the remaining 37 participants.

### Experimental design

Stimulus presentation and behavioral data collection were performed using the SMILE package (1.0.0).

**Attention task.** The design of the current study was based on the 'Central Expectancy Task' reported in ref. 34. To reduce exogenous attention, a central diamond and two flanking squares on either side were present throughout the session and participants were instructed to maintain central fixation throughout the study (Fig. 1A). There were a total of 1152 trials divided into 16 runs. Each trial began with a cue, a centrally presented single- or double-headed arrow. There were three cue types, valid, invalid, and neutral. Participants were instructed to covertly attend to the cued (valid/invalid) location on trials with a single-headed arrow. 66% of cues were valid or invalid (n = 768) and 33% were neutral (n = 384). 75% of cued trials were validly cued (n = 576) and 25% of cued trials were invalidly cued (n = 192). The cue remained on the screen for 100 ms. After a variable stimulus onset asynchrony (SOA), a probe stimulus appeared. The three SOA conditions were 200, 400, and 800 ms. The probe stimulus could either be a cross or a plus and the probe remained on the screen for 100 ms. Participants were to respond by pressing a keyboard key when they detected a cross probe and to withhold a response when they detected a plus probe. Participants were encouraged to respond as quickly and accurately as possible.

Participants were given a 500 ms interval to respond to the probe. Each trial was separated by a 500 ms interstimulus interval. Trial condition varied randomly; cue type, SOA and probe type were fully crossed and conditions were distributed equally within each run.

**Mnemonic state task.** Participants were biased via explicit instructions on a trial-by-trial basis to engage an encoding or retrieval state, while perceptual input and behavioral demands were held constant. In this mnemonic state task (for specific study parameters, please see refs. [11], [12]), participants viewed two lists of object images. For the first list, each object was new. For the second list, each object was again new but was categorically related to an object from the first list. For example, if List 1 contained an image of a bench, List 2 would contain an image of a different bench. During List 1, participants were instructed to encode each new object. During List 2, however, each trial contained an instruction to either encode the current object (e.g., the new bench) or to retrieve the corresponding object from List 1 (the old bench). Each object was presented for 2000 ms. Participants completed either a two-alternative forced choice recognition test or a recency test on the object stimuli. I used the stimulus-locked List 2 data to train a multivariate pattern classifier (see below) to distinguish encoding and retrieval states.

## EEG data acquisition and preprocessing

EEG recordings were collected using a BrainVision system and an ActiCap equipped with 64 Ag/AgCl active electrodes positioned according to the extended 10–20 system. All electrodes were digitized at a sampling rate of 1000 Hz and were referenced to electrode FCz. Offline, electrodes were later converted to an average reference. Impedances of all electrodes were kept below 50 kΩ. Electrodes that demonstrated high impedance or poor contact with the scalp were excluded from the average reference. Bad electrodes were determined by voltage thresholding (see below).

Custom Python codes were used to process the EEG data. I applied a high pass filter at 0.1 Hz, followed by a notch filter at 60 Hz and harmonics of 60 Hz to each participant's raw EEG data. I then performed three preprocessing steps[59] to identify electrodes with severe artifacts. First, I calculated the mean correlation between each electrode and all other electrodes as electrodes should be moderately correlated with other electrodes due to volume conduction. I z-scored these means across electrodes and rejected electrodes with z-scores <−3. Second, I calculated the variance for each electrode as electrodes with very high or low variance across a session are likely dominated by noise or have poor contact with the scalp. I then z-scored variance across electrodes and rejected electrodes with a $|z| \geq 3$. Finally, I expect many electrical signals to be autocorrelated, but signals generated by the brain versus noise are likely to have different forms of autocorrelation. Therefore, I calculated the Hurst exponent, a measure of long-range autocorrelation, for each electrode and rejected electrodes with a $|z| \geq 3$. Electrodes marked as bad by this procedure were excluded from the average re-reference. I then calculated the average voltage across all remaining electrodes at each time sample and re-referenced the data by subtracting the average voltage from the filtered EEG data. I used wavelet-enhanced independent component analysis[60] to remove artifacts from eyeblinks and saccades.

**EEG data analysis.** For the attention task data, I applied the Morlet wavelet transform (wave number 6) to all electrode EEG signals from 500 ms preceding to 2000 ms following cue onset, across 46 logarithmically spaced frequencies (2–100 Hz[61]). After log-transforming the power, I downsampled the data by taking a moving average across 100 ms time windows and sliding the window every 25 ms, resulting in 97 time windows (25 non-overlapping). Power values were then z-transformed by subtracting the mean and dividing by the standard deviation power. Mean and standard deviation power were calculated across all trials and across time points for each frequency. I followed the same procedure for the mnemonic state task, with 317 overlapping (80 non-overlapping) time windows from 4000 ms preceding to 4000 ms following stimulus onset[11].

**Pattern classification analyses.** Pattern classification analyses were performed using penalized (L2) logistic regression implemented via the sklearn module (0.24.2) in Python and custom Python code. For all classification analyses, classifier features were comprised of spectral power across 63 electrodes and 46 frequencies. Before pattern classification analyses were performed, an additional round of z-scoring was performed across features (electrodes and frequencies) to eliminate trial-level differences in spectral power[11,62,63]. Therefore, mean univariate activity was matched precisely across all conditions and trial types. I assess classifier performance via classification accuracy and classifier evidence. Classification accuracy reflects a binary coding of whether the classifier correctly guessed a condition label (e.g. cue direction, left/right/neutral). Classifier evidence is a continuous value reflecting the logit-transformed probability that the classifier assigned the correct condition label. I used classification accuracy for general assessment of how well cue and probe information could be decoded. I used classifier evidence as a trial-specific, continuous measure of information about the cue or probe, which I related to trial-level retrieval state evidence.

**Cue direction classification.** I conducted within participant leave-one-run-out cross-validated classification (penalty parameter = 1). Each classifier (one per SOA condition, three in total) was trained to distinguish left, right, and neutral cue trials based on spectral power averaged over the delay interval (100–300 ms, 100–500 ms, or 100–900 ms respectively). I used classifier evidence for the neutral cue as a baseline, and subtracted these values from evidence for either the left or right cue depending on the actual cue direction.

**Probe location and identity classification.** I conducted within participant leave-one-run-out cross-validated classification (penalty parameter = 1). To test the extent to which probe location and identity could be decoded, I trained a classifier to distinguish either left vs. right presented probe trials or cross vs. plus probe trials based on spectral power averaged across the response interval (0–500 ms). To relate probe information to the retrieval state, I performed classification across each of the five time windows in the response interval and correlated trial-level location or identity evidence with retrieval state evidence.

**Cross study memory state classification.** To measure retrieval state engagement in the attention task, I conducted three stages of classification. First, I conducted within participant leave-one-run-out cross-validated classification (penalty parameter = 1) on all participants who completed the mnemonic state task ($N = 100$, see ref. [12] for details). The classifier was trained to distinguish encode vs. retrieve states based on spectral power averaged across the 2000 ms stimulus interval during List 2 trials. I utilized the full stimulus interval as I have previously found robust within-participant decoding during this interval[9] and did not have a priori predictions regarding which time interval(s) would have the strongest memory state dissociations. For each participant, I generated true and null classification accuracy values. I permuted condition labels (encode, retrieve) for 1000 iterations to generate a null distribution for each participant. Any participant whose true classification accuracy fell above the 90th percentile of their respective null distribution was selected for further analysis ($N = 35$). The proportion of above threshold participants is in line with my prior work[9]. Participants who fall below threshold may not consistently engage encoding and retrieval states throughout the entire stimulus interval, although this does not necessarily mean that they do not engage the same neural mechanisms as the above-threshold

participants. All of the mnemonic state classifier dependent analyses produce qualitatively similar results when the full dataset of $N = 100$ participants is used to train the classifier (see Supplementary Information). Second, I conducted leave-one-participant-out cross-validated classification (penalty parameter = 0.0001) on the selected participants to validate the mnemonic state classifier. I found significantly above chance classification accuracy ($M = 60\%$, SD = 9.5%, $t_{34} = 6.1507$, $p < 0.0001$, $d = 1.501$, CI = [0.069,0.1371]; Fig. 2A), indicating that the cross participant classifier is able to distinguish encoding and retrieval states. Of these 35 participants, 13 also completed the attention task. Finally, I applied the cross participant mnemonic state classifier to the attention task data, specifically spectral signals in 100 ms time windows. I extracted classifier evidence, the logit-transformed probability that the classifier assigned a given attention task trial a label of encoding or retrieval. This approach provides a trial-level estimate of retrieval state evidence during the attention task.

**Statistical analyses.** To assess the impact of cue type and SOA on behavior, I performed a repeated measures ANOVA on reaction times. I used repeated measures ANOVAs (rmANOVAs) to assess the impact of cue type, SOA, probe type, and time on retrieval state evidence. I followed-up significant interactions with post-hoc paired $t$-tests.

I used paired-sample $t$-tests to compare classification accuracy across participants to chance decoding accuracy, as determined by permutation procedures. Namely, for each participant, I shuffled the condition labels of interest (e.g., left and right for the probe location classifier) and then calculated classification accuracy. I repeated this procedure 1000 times for each participant and then averaged the 1000 shuffled accuracy values for each participant. These mean values were used as participant-specific empirically derived measures of chance accuracy.

I used Pearson correlations to relate trial-level retrieval evidence to trial-level cue direction evidence, probe location evidence, and probe identity evidence. I Fisher-Z transformed all resulting rho values. I used one sample $t$-tests to compare participant-level zRho values to zero.

I used a general linear model to predict trial-level reaction times (RTs) based on retrieval evidence and probe identity evidence. The only included trials were hits, cross targets to which participants responded. I used one sample $t$-tests to compare participant-level beta values to zero.

I used false discovery rate (FDR) to correct for multiple comparisons[64] for both post-hoc ANOVAs and post-hoc $t$-tests. I report effect sizes as partial eta squared ($\eta_p^2$) and Cohen's $d$.

To perform Bayes Factor analyses, I used the BayesFactor package (version 0.9.12-4.4) in R (version 4.2.3) with the default prior settings and specifically the linear model function lmBF to compare models.

### Reporting summary

Further information on research design is available in the Nature Portfolio Reporting Summary linked to this article.

## Data availability

The raw data generated in this study have been deposited in the Open Science Foundation database at https://doi.org/10.17605/OSF.IO/DU56C. All data necessary to reproduce the results, including all display items, have been shared in the OSF repository.

## Code availability

All experimental codes used for data collection and all analysis codes used for data analysis have been deposited in the Open Science Foundation database at https://doi.org/10.17605/OSF.IO/DU56C. All code necessary to reproduce the results, including all display items, have been shared in the OSF repository.

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

## Acknowledgements

I thank Yuju Hong for assistance with data collection.

## Author contributions

N.M.L. conceptualized the work, collected and analyzed the data, and wrote the manuscript.

## Competing interests

The author declares no competing interests.
