## [Peer Review File · Nature Communications]

Reviewers' Comments:

Reviewer #1:

Remarks to the Author:

This study investigates whether a memory retrieval state can be conceptualized as an internal attention state. Participants perform an attention task. A classifier trained on an independent dataset to dissociate encoding vs retrieval states is used to assess whether the attention task can be classified as retrieval. In the attention task, a cue points towards one of the two locations. After a while, a probe arrives either at the cued or the uncued location. On neutral trials, there is no cue. The probe can be the target, which requires a response, or a lure, which requires no response. Critically, participants attend to the cued location until the probe and also internally attend to the mental representation of the probe to decide whether to respond or not.

The retrieval state classifier gives above-chance decoding, particularly in cue (vs neutral) trials and at longer delays. The retrieval evidence increases towards the response in the attention task and is correlated with faster responses. Based on these results, the author suggests that a retrieval state may simply be an internal attention state.

I think this is a neat, elegant, and timely paper. The results are mostly convincing. However, the conclusions are rather too strong. Below, I describe some aspects that can be improved.

1) In my opinion, the major weakness of the paper is how the results are interpreted. A classifier trained to decode retrieval can *involve* internal attention mechanisms, which can explain cross-task classification. However, this is not enough to conclude that a retrieval state is *akin to* an internal attention state. Retrieval can involve internal attention but this does not necessarily mean it can be reconceptualized as internal attention.

The strictest test of a complete overlap of retrieval and internal attention would be to decode the particular task participants are performing. If a classifier can tell apart these two tasks, then the tasks must have some differences. Alternatively, the paper can tone down its arguments. Instead of conceptualizing retrieval as internal attention, it can argue that internal attention is an integral part of retrieval. A third option could be to make use of a modelling approach in an attempt to estimate the degree of overlap between retrieval and internal attention, though I do not have a precise suggestion on how to do this. In any case, I think the current interpretation of the data is a bit extreme.

2) There was no difference between neutral and invalid in retrieval classification. These two conditions both require a reorienting of attention to the correct target location, as also stated by the author. The author proposes two possible explanations; i) less external attention (hence more internal attention) to process the cue on valid trials, or ii) no need to reorient attention allowing participants to engage the retrieval state earlier. I find the 2nd argument circular. In most of the paper, the author argues that the present task provides retrieval evidence because of the shared internal attention demands of the present task and the retrieval task used to train the classifier. Instead, in this section (Page 8, end of the 2nd paragraph) the author suggests that the retrieval evidence reflects retrieval itself. If so, then isn't it counter to the main conclusion of the paper?

An alternative could be that the classifier prefers internal attention to memory representations that have been encoded earlier (eg. the cued location) over newly encountered representations (eg. the probe location). More specifically, internal attention to active/online (working) memories can differ from internal attention to memories retrieved from a passive/offline (longer-term) memory state. Exploring the similarities and differences between internal attention to different memory types can potentially increase the impact of the paper.

3) The specific type of attention described in the paper can be more specific. There are different types of attention (sustained, selective, object-based, spatial, endogenous, etc) and it is unclear which is suggested to underlie the retrieval state.

The author does mention that a retrieval state reflects any form of internally directed attention. However, in my opinion, the type of attention that both retrieval and the attention task in the present study is mainly restricted to selective attention. If the author agrees, this can be clarified more in the paper. On the other hand, if the author does not agree with this, then discussing why

so in the Discussion section would also be useful.

4) On a similar note, it is not clear based on the present data that attention is not spatial in nature. As the author argues, the classifier is not based on attending to a particular location. However, the classifier might be reflecting spatially attending to any internal location. For example, in the original retrieval task, participants can be retrieving the information at the location at which the memory item was encoded. The fact that objects were always centrally presented does not preclude this possibility, especially given that memories tend to be bound to particular locations even when location is task-irrelevant (e.g., Foster et al., 2017; Tam & Wyble, 2022). Therefore, both the memory task used to train the classifier and the attention task used to test the classifier might have a spatial selective attention component in common that drives successful classification. Once again, in my opinion, discussing what type of internal attention is the driving factor behind successful decoding is important.

Foster, J. J., Bsales, E. M., Jaffe, R. J., & Awh, E. (2017). Alpha-band activity reveals spontaneous representations of spatial position in visual working memory. *Current Biology*, 27(20), 3216-3223.

Tam, J., & Wyble, B. (2022). Location has a privilege, but it is limited: Evidence from probing task-irrelevant location. *Journal of Experimental Psychology: Learning, Memory, and Cognition*.

5) A recent paper found that the same inverted encoding model can reconstruct information stored in working memory and retrieved from long-term memory. I think that given the overlap between attention and working memory, the results of the present study can be discussed concerning this study (Vo et al., 2022). Given that the algorithm in Vo et al. (2022) is based on specific spatial content while the one in the present paper does not, the present work makes a novel contribution beyond Vo et al. (2022) that can be further discussed in the Discussion if the author thinks this is useful.

6) On a related note, I believe that the discussion on the intermediary role of working memory in connecting internal attention and retrieval deserved deeper discussion. First, internal attention is critical for working memory. Second, working memory has been suggested to represent information retrieved from long-term memory (Fukuda et al., 2017; Vo et al., 2022). Thus, internal attention may be part of retrieval because the retrieved content is maintained in working memory.

Fukuda, K., & Woodman, G. F. (2017). Visual working memory buffers information retrieved from visual long-term memory. *Proceedings of the National Academy of Sciences*, 114(20), 5306-5311.

Vo, V. A., Sutterer, D. W., Foster, J. J., Sprague, T. C., Awh, E., & Serences, J. T. (2022). Shared Representational Formats for Information Maintained in Working Memory and Information Retrieved from Long-Term Memory. *Cerebral Cortex*, 32(5), 1077-1092.

7) The paper states that the accuracy of decoding the retrieval state fell above the 90th percentile of the null distribution in only 35 out of 100 participants in the original study. This can raise doubts in some readers. Therefore, the author can discuss why decoding the retrieval state may be so weak/unsuccessful. Is it possible that the original decoder can decode retrieval only if participants are strongly internally attending? This can partly explain the cross-task classification with the attention task.

Best regards,
Eren Günseli

Reviewer #2:

Remarks to the Author:

Summary: EEG data were collected while observers completed a memory encoding and retrieval task, and a pattern classifier was trained to distinguish between encoding and retrieval trials. The training weights obtained from a subset of participants that completed the encoding vs retrieval task were then applied to EEG data recorded while observers completed a Posner style

endogenous cueing task. Classifier evidence revealed that patterns of EEG activity during the attention task more closely resembled retrieval trials than encoding trials. The manuscript concludes that the memory retrieval state constitutes internal attention.

Evaluation: I really enjoyed reading and thinking about this paper, and think that questions about the extent that memory retrieval and attention overlap is of broad interest to the field. Additionally, I thought the analytic approach of training a pattern classifier on one task, and then asking how much brain activity during a separate task represented one mental state or another was novel and clever.

Despite my enthusiasm for many aspects of the paper, I'm not convinced that the results reported here support the broad conclusions drawn in the manuscript. Specifically, while it is intriguing and even surprising that EEG data during the attention task more closely resembles memory retrieval than encoding, I'm not convinced that we can conclude that internal attention explains the relationship. I also find it problematic that it was only possible to successfully classify retrieval vs encoding state in 35 of the 100 training observers. I've outlined these comments and a few others in more detail below:

Primary comments:

1. As the manuscript highlights, there are many cognitive operations involved in both the attention and retrieval tasks. Unfortunately this feature of the task makes it difficult to conclude what cognitive process or processes underlies the observed similarities. I agree that both tasks rely on internal attention/working memory in some capacity. However, engaging in any computer based task also requires observers to direct their external attention to the computer display while they complete the task, even when the display is blank aside from fixation. Thus, a plausible alternative explanation is that observers are simply more likely to broaden their spotlight of external attention (or move the spotlight away from fixation) in the retrieval and attention tasks compared to the encoding task when they are more closely monitoring a shape at fixation.
2. Since it was only possible to distinguish between encoding and retrieval states for $\sim\frac{1}{3}$ of observers, it doesn't seem like the encoding and retrieval task is a good candidate for measuring the relationship between retrieval and attention in the general population. That said, I might be missing something here. This concern could be alleviated by demonstrating that the same pattern of results is observed when all encoding and retrieval observers are included.
3. The current results tell us that the brain activity during the attention task is more like retrieval trials than encoding trials, but doesn't tell us how similar activity is between the attention and retrieval task. One way to test this, is to conduct a cross subject decoding analysis that includes all three tasks. An observation of equal retrieval and attention classifier evidence for retrieval and attention trials, would provide strong evidence for the conclusion that both tasks rely on the same cognitive process.
4. The reported analyses always train and test on the same time points (i.e., train at 100 ms in the encoding/retrieval task and test at 100 ms in the attention task). Since the tasks themselves are so different, matched time points may not provide the best decoding. A straightforward way to test this would be to include a temporal generalizability analysis where each time point of the training task is used to classify each time point of the testing task (see MacLean et al., 2019 figure 3 for a nice example).
5. Interpreting the cross-task decoding results requires a clear understanding of the encoding vs retrieval decoding performance, so it would be helpful to include a plot of the time-resolved encoding vs retrieval classification results.

References:

MacLean MH, Bullock T, Giesbrecht B (2019) Dual Process Coding of Recalled Locations in Human Oscillatory Brain Activity. *The Journal of Neuroscience* 39:6737.

Reviewer #3:

Remarks to the Author:

Review of "Reconceptualizing the retrieval state as an internal attention state"

This paper tests the hypothesis that "retrieval state" (i.e., whether you use a stimulus as a prompt to retrieve a related memory, vs. encoding the stimulus as a new memory) can be conceptualized as internally-directed attention. The author's strategy for testing this hypothesis is to take a "retrieval state" EEG classifier (trained on an already-collected dataset), apply it to Posner-like cued spatial attention task, and then assess whether the output of the classifier aligns with how we think internally-directed attention would behave in that task. Essentially, the approach here is to build an extremely detailed "circumstantial" case for the equivalence between the processes – the paper analyzes how retrieval state evidence varies as a function of a wide range of factors (e.g., stimulus onset asynchrony, cue type) and how it relates to other kinds of classifier evidence (e.g., for cue direction, probe location, and probe identity) across all of the different time epochs of the trial, and then argues how each of these findings is what you would expect if retrieval state evidence were tracking internally-directed attention. The case that the author builds is very thorough and the total weight of evidence in support of the author's argument is strong; while it's not clear if the two processes are exactly the same (see paragraph below), this paper shows that, at the very least, they are quite closely related. As such, I think it's an important contribution. I also am a big fan of the approach of taking a pre-trained classifier from another task and applying it to a new task – as illustrated by the present study, this process will allow our field to get an increasingly refined sense of what the classifier measures. Over time, I can imagine the field accumulating a "library" of well-validated classifiers that each tap into distinct processes.

My only substantive comment relates to the challenges inherent in establishing the "identity" of retrieval state and internally-directed attention. As noted above, the two processes clearly (from the results shown here) have many elements in common, but the approach taken in this paper can not rule out the possibility that there are some subprocesses related to retrieval state that are not related to internally-directed attention, or vice-versa. In an ideal world, one could have multiple tasks tracking retrieval state (say, A and B) and multiple tasks tracking internally-directed attention (say, C and D) and then show that all possible pairs of tasks cross-generalize equally well (e.g., training on a retrieval state task generalizes to internally-directed attention task C as well as training on internally-directed attention task D generalizes to internally-directed attention task C; and it would also be useful to show that training on an internally-generated attention task generalizes to retrieval state task A as well as training on retrieval state task B generalizes to retrieval state task A). Having said this, I don't think it's reasonable to expect the author to do something like this in the scope of this paper; in my view, the contribution is strong enough as it stands. Consequently, I would be satisfied with the author adding a few caveats about the kinds of inferences about process identity that are licensed by the design they used.

Sincerely,

Ken Norman (I sign all of my reviews)

March 2, 2023

Thank you for the opportunity to revise the manuscript, “Reconceptualizing the retrieval state as an internal attention state.” I appreciate the Reviewers’ helpful suggestions and comments. I have made thorough revisions to the manuscript to address these comments and I believe that the manuscript has been substantially strengthened. I have provided an item-by-item summary of the revisions to the Reviewers’ concerns (italicized and in blue; responses in black) in the pages below.

Reviewer #1 (Remarks to the Author):

This study investigates whether a memory retrieval state can be conceptualized as an internal attention state. Participants perform an attention task. A classifier trained on an independent dataset to dissociate encoding vs retrieval states is used to assess whether the attention task can be classified as retrieval. In the attention task, a cue points towards one of the two locations. After a while, a probe arrives either at the cued or the uncued location. On neutral trials, there is no cue. The probe can be the target, which requires a response, or a lure, which requires no response. Critically, participants attend to the cued location until the probe and also internally attend to the mental representation of the probe to decide whether to respond or not.

The retrieval state classifier gives above-chance decoding, particularly in cue (vs neutral) trials and at longer delays. The retrieval evidence increases towards the response in the attention task and is correlated with faster responses. Based on these results, the author suggests that a retrieval state may simply be an internal attention state.

I think this is a neat, elegant, and timely paper. The results are mostly convincing. However, the conclusions are rather too strong. Below, I describe some aspects that can be improved.

I thank the Reviewer for their helpful comments and positive feedback on the manuscript. I have revised the manuscript to address the Reviewer's concern regarding the interpretations and provide item-by-item responses below.

- 1. In my opinion, the major weakness of the paper is how the results are interpreted. A classifier trained to decode retrieval can involve internal attention mechanisms, which can explain cross-task classification. However, this is not enough to conclude that a retrieval state is akin to an internal attention state. Retrieval can involve internal attention but this does not necessarily mean it can be reconceptualized as internal attention. The strictest test of a complete overlap of retrieval and internal attention would be to decode the particular task participants are performing. If a classifier can tell apart these two tasks, then the tasks must have some differences. Alternatively, the paper can tone down its arguments. Instead of conceptualizing retrieval as internal attention, it can argue that internal attention is an integral part of retrieval. A third option could be to make use of a modelling approach in an attempt to estimate the degree of overlap between retrieval and internal attention, though I do not have a precise suggestion on how to do this. In any case, I think the current interpretation of the data is a bit extreme.*

I thank all three Reviewers for raising this very important point and I agree that the strictest test of the overlap between retrieval and internal attention is through cross-task classification. The current study is not ideally suited for cross-task decoding of encoding vs. retrieval vs. attention given that the mnemonic data are within subject (the same participants performed encoding and retrieval) whereas the cross-study application is between subject. Additionally, to the extent that such a task exists, the attention task in the current manuscript is not a "pure" internal attention task. Instead, attentional demands are expected to change between external (to cue and probe) and internal (to maintained information). The attention task conditions – cue type and delay – also impact the degree and temporal dynamics of attention allocation, meaning that attention is likely to vary across time and trials in the attention task. Together, as laid out by Reviewer 3, the strongest approach would be to utilize multiple specific retrieval and internal attention tasks and demonstrate successful cross-task classification which is best suited to future work. Therefore, I have toned down the interpretation of the current data by making edits throughout the text (including the title), and specifically address this limitation and potential future direction in the discussion (page 16), text copied below.

A critical next step will be to perform direct cross-task classification on a set of a retrieval and internal attention tasks in order to determine the extent to which the two tasks overlap. Robust cross-task classification would provide strong evidence that the retrieval state and internal attention are one and the same, whereas the current findings leave open the possibility that specific processes beyond internal attention are unique to the retrieval state.

2. *There was no difference between neutral and invalid in retrieval classification. These two conditions both require a reorienting of attention to the correct target location, as also stated by the author. The author proposes two possible explanations; i) less external attention (hence more internal attention) to process the cue on valid trials, or ii) no need to reorient attention allowing participants to engage the retrieval state earlier. I find the 2nd argument circular. In most of the paper, the author argues that the present task provides retrieval evidence because of the shared internal attention demands of the present task and the retrieval task used to train the classifier. Instead, in this section (Page 8, end of the 2nd paragraph) the author suggests that the retrieval evidence reflects retrieval itself. If so, then isn't it counter to the main conclusion of the paper?*

An alternative could be that the classifier prefers internal attention to memory representations that have been encoded earlier (eg. the cued location) over newly encountered representations (eg. the probe location). More specifically, internal attention to active/online (working) memories can differ from internal attention to memories retrieved from a passive/offline (longer-term) memory state. Exploring the similarities and differences between internal attention to different memory types can potentially increase the impact of the paper.

I understand the Reviewer's concern and realize that I did not clearly articulate my second point. I do not think that on validly cued trials participants are "retrieving" – in the classic sense – any prior information or stimuli. Instead, I view *switching* between external vs. internal attention and *reorienting* external attention to a different spatial location as two potentially distinct processes, either or both of which may influence the current estimates of retrieval state evidence. Valid trials reduce demands to switch from an internal to an external state (to process the probe) and reduce or eliminate demands to reorient external attention from one spatial location to another. In contrast, neutral and/or invalid trials may require either switching from internal to external attention and/or reorienting external attention to a different spatial location. Decreased retrieval evidence during trials with short SOAs and/or valid cues could be reflective of either internal to external switching or spatial location reorienting. I have modified the text in the results (page 8) to clarify these points and have copied the text below.

These results show that valid cues facilitate entrance into a retrieval state. It may be that less external attention is needed to process the probe or that no reorientation of external attention (to a different location) is needed on valid trials. Reduced demand to either switch into an external state and/or to reorient external attention to a different spatial location may result in the observed retrieval state evidence on validly cued trials.

The Reviewer raises an interesting point regarding the recency of memory representations and the mnemonic state classifier. Given that retrieval state evidence correlates with both cue direction and probe location information, it does not appear that there is a classifier preference for older vs. newer encountered representations, at least on the timescale of milliseconds. In prior work (Smith et al., 2022), my lab has found that memory states generalize across objects presented near vs. far in time (operationalized as the difference in serial position of the object pairs), suggesting that the classifier does not have a preference for earlier vs. more recently encountered stimuli. Furthermore, given

the study by Vo and colleagues cited below in which reinstatement is largely similar across working memory maintenance and long term memory retrieval, I would anticipate similar recruitment of internal attention for maintenance of both types of memories. I expand on this below in response to the Reviewer's 5th and 6th comments. I have incorporated consideration of this point (along with further consideration of the role of working memory) in the discussion (pages 14-15); I've copied the text below.

Working memory maintenance may serve as the mechanism that links the retrieval state and internal attention. There is an intimate connection between selective attention, working memory, and long term memory¹⁶. In particular, selective attention and working memory processes recruit shared neural substrates^{17,39,45,47}. Furthermore, univariate working memory maintenance signals are engaged during long term memory retrieval⁴⁸ and multivariate perceptual representations are reinstated both during working memory maintenance and long term memory retrieval⁴⁹. Taken together, these findings support the interpretation that working memory maintenance may support maintenance of both the retrieved stimuli in the mnemonic state task and the cue/probe spatial information in the attention task. A prediction from this interpretation is that we should observe retrieval state evidence during working memory recruitment.

3. *The specific type of attention described in the paper can be more specific. There are different types of attention (sustained, selective, object-based, spatial, endogenous, etc) and it is unclear which is suggested to underlie the retrieval state.*

The author does mention that a retrieval state reflects any form of internally directed attention. However, in my opinion, the type of attention that both retrieval and the attention task in the present study is mainly restricted to selective attention. If the author agrees, this can be clarified more in the paper. On the other hand, if the author does not agree with this, then discussing why so in the Discussion section would also be useful.

I thank the Reviewer for raising this point and I agree that the present study is indexing selective attention whereby participants are selecting an internal stimulus representation for processing in the retrieval task or selecting a spatial location in the attention task. I have modified text in the introduction (page 2) and discussion (pages 13, 14) to address this point, text copied below.

Internal attention is the selection of stored representations and lies in contrast to external attention, the selection of sensory stimuli¹⁵. (Page 2)

Internal attention is the selection of the stored contents of the mind, including working and long term memory representations¹⁵. (Page 13)

Our interpretation is that participants direct their mind's eye inwards at multiple points in a given trial in the attention task and that retrieval evidence tracks the extent to which participants have selected internal representations. (Page 14)

Internal attention may still be necessary for successful episodic retrieval – the original proposed function of the retrieval mode – but internal attention will also be necessary for selecting any form of internal information, regardless of the specific content. (Page 14)

Beyond fluctuating over time, we have directly linked the retrieval state to the information to internal information that is selected. (Page 14)

4. *On a similar note, it is not clear based on the present data that attention is not spatial in nature. As the author argues, the classifier is not based on attending to a particular location. However, the classifier might be reflecting spatially attending to any internal location. For example, in the original retrieval task, participants can be retrieving the information at the location at which the memory item was encoded. The fact that objects were always centrally presented does not preclude this possibility, especially given that memories tend to be bound to particular locations even when location is task-irrelevant (e.g., Foster et al., 2017; Tam & Wyble, 2022). Therefore, both the memory task used to train the classifier and the attention task used to test the classifier might have a spatial selective attention component in common that drives successful classification. Once again, in my opinion, discussing what type of internal attention is the driving factor behind successful decoding is important.*

*Foster, J. J., Bsaies, E. M., Jaffe, R. J., & Awh, E. (2017). Alpha-band activity reveals spontaneous representations of spatial position in visual working memory. *Current Biology*, 27(20), 3216-3223.*

*Tam, J., & Wyble, B. (2022). Location has a privilege, but it is limited: Evidence from probing task-irrelevant location. *Journal of Experimental Psychology: Learning, Memory, and Cognition*.*

Although the possibility that retrieval evidence is indexing attention to a spatial location cannot be ruled out, I would be hesitant to draw the strong conclusion that retrieval evidence specifically reflects attending to a particular spatial location. First, due to the nature of the attention task, the majority of maintained information is spatial, meaning that there are few direct comparisons to other information that could be maintained. Second, a critical difference between the studies referenced by the Reviewer and the current study is the diagnosticity of spatial location. Although it is the case that participants appear to encode/retrieve spatial information when it is task-irrelevant, in all of these studies the stimuli have variable spatial locations, meaning that spatial location is potentially diagnostic – it can be used to categorize and/or identify stimuli. In contrast, in the mnemonic state task with only one location for all stimuli, spatial location is non-diagnostic. Whereas diagnostic features are encoded/retrieved even when task-irrelevant (Hsu et al., 2014, *Journal of Cognitive Neuroscience*), broadly speaking, neuronal representation is generally weaker for non-diagnostic features (Sigala & Logothetis, 2002, *Nature*). It thus seems unlikely that participants are encoding the spatial location information in the mnemonic state task. Finally, even if participants do encode/retrieve location information in the mnemonic state task, it is challenging to envision how the classifier would have access to central location information in this task. Every trial contains a centrally presented object stimulus and a retrievable centrally presented object stimulus, thus whether the participant engages an encoding or a retrieval state, the same central location information will be present. Given that the classifier can distinguish encode and retrieve trials, it seems unlikely that such dissociation is based on location information; however, a study that directly tests whether retrieval evidence is modulated by the maintenance of non-spatial information is needed to test this interpretation. I've expanded on this point in the discussion (page 15) and have copied the text below.

Although location information could be automatically encoded or retrieved⁴⁹, this spatial information would not differentiate encode and retrieve trials and therefore is unlikely to be driving classifier performance. Furthermore, the classifier had no *a priori* information about left vs. right spatial locations and thus cannot purely reflect such information. Instead, our interpretation is that the retrieval state reflects any form of internally directed attention, whether

to spatial, episodic, semantic, or other perceptual information that is not currently present in the external environment. However, future work is necessary to directly test the extent to which the retrieval state is driven by maintenance of spatial vs. non-spatial information.

5. *A recent paper found that the same inverted encoding model can reconstruct information stored in working memory and retrieved from long-term memory. I think that given the overlap between attention and working memory, the results of the present study can be discussed concerning this study (Vo et al., 2022). Given that the algorithm in Vo et al. (2022) is based on specific spatial content while the one in the present paper does not, the present work makes a novel contribution beyond Vo et al. (2022) that can be further discussed in the Discussion if the author thinks this is useful.*

I agree with the Reviewer that the Vo et al. 2022 paper is highly relevant to the interpretation that working memory may be the mechanism connecting the retrieval state and internal attention. I now include the Vo et al. paper in the discussion (pages 14-15), text copied below.

Furthermore, univariate working memory maintenance signals are engaged during long term memory retrieval⁴⁸ and multivariate perceptual representations are reinstated both during working memory maintenance and long term memory retrieval⁴⁹.

6. *On a related note, I believe that the discussion on the intermediary role of working memory in connecting internal attention and retrieval deserved deeper discussion. First, internal attention is critical for working memory. Second, working memory has been suggested to represent information retrieved from long-term memory (Fukuda et al., 2017; Vo et al., 2022). Thus, internal attention may be part of retrieval because the retrieved content is maintained in working memory.*

*Fukuda, K., & Woodman, G. F. (2017). Visual working memory buffers information retrieved from visual long-term memory. *Proceedings of the National Academy of Sciences*, 114(20), 5306-5311.*

*Vo, V. A., Sutterer, D. W., Foster, J. J., Sprague, T. C., Awh, E., & Serences, J. T. (2022). Shared Representational Formats for Information Maintained in Working Memory and Information Retrieved from Long-Term Memory. *Cerebral Cortex*, 32(5), 1077-1092.*

I agree with the Reviewer that further discussion of the intermediary role of working memory would bolster the manuscript. Indeed, it seems very plausible that the commonality between the two tasks may be explained by internal attention directed to retrieved stimuli in the mnemonic task and internal attention directed to the cue/probe information in the attention task. I now address this point in the discussion (pages 14-15), text copied below.

Working memory maintenance may serve as the mechanism that links the retrieval state and internal attention. There is an intimate connection between selective attention, working memory, and long term memory¹⁶. In particular, selective attention and working memory processes recruit shared neural substrates^{17, 39, 45-47}. Furthermore, univariate working memory maintenance signals are engaged during long term memory retrieval⁴⁸ and multivariate perceptual representations are reinstated both during working memory maintenance and long term memory retrieval⁴⁹. Taken together, these findings support the interpretation that working memory may be recruited for maintenance of both the retrieved stimuli in the mnemonic state task and the cue/probe spatial information in the attention task. A prediction from this

interpretation is that retrieval state engagement should be observed in any task which includes/relies on working memory maintenance.

7. *The paper states that the accuracy of decoding the retrieval state fell above the 90th percentile of the null distribution in only 35 out of 100 participants in the original study. This can raise doubts in some readers. Therefore, the author can discuss why decoding the retrieval state may be so weak/unsuccessful. Is it possible that the original decoder can decode retrieval only if participants are strongly internally attending? This can partly explain the cross-task classification with the attention task.*

Best regards,

Eren Günseli

I thank the Reviewer for raising this point, which was also raised by Reviewer 2. There are several potential explanations as to why only one third of participants exceed the threshold for inclusion in the cross-subject classifier. First, noisier data in general could interfere with classification and some participants may have noisier EEG signals than others. Second, participants could fail to follow instructions and not engage encoding or retrieval. As it is common practice to exclude participants with poor data quality and poor behavioral performance – and such participants were excluded prior to the aggregation of the full training set of N=100 – these accounts are less likely, though of course not impossible. A more interesting alternative is that some participants may not engage memory states in the anticipated manner – i.e. solely engaging encoding when instructed to encode and solely engaging retrieval when instructed to retrieve. This may include the extent to which participants engage internally directed attention. As every trial has an externally presented stimulus that could be encoded and a categorically related stimulus that could be retrieved, participants could switch between encoding and retrieval during a single trial, preventing the classifier from both learning from that trial and correctly labeling it. Indeed, my lab has previously found that participants may automatically engage retrieval even when instructed to encode, before “correcting” and switching into a retrieval state (Smith et al., 2022). Investigations of memory state switching is outside of the scope of the present work, but may account for the variability in within-subject classification accuracy.

The proportion of participants that exceeded the 90th percentile threshold is consistent with prior work that I have conducted using the same mnemonic state task in an independent set of participants (Long & Kuhl, 2019). I have added text to the methods (page 22) indicating that the proportion of above-threshold participants is in line with prior findings and that below-threshold participants may inconsistently recruit the same mechanisms as above-threshold participants.

The proportion of above threshold participants is in line with our prior work⁹. Participants who fall below threshold may not consistently engage encoding and retrieval states throughout the entire stimulus interval, although this does not necessarily mean that they do not engage the same neural mechanisms as the above-threshold participants.

Reviewer #2 (Remarks to the Author):

Summary: EEG data were collected while observers completed a memory encoding and retrieval task, and a pattern classifier was trained to distinguish between encoding and retrieval trials. The training weights obtained from a subset of participants that completed the encoding vs retrieval task were then applied to EEG data recorded while observers completed a Posner style endogenous cueing task. Classifier evidence revealed that patterns of EEG activity during the attention task more closely resembled retrieval trials than encoding trials. The manuscript concludes that the memory retrieval state constitutes internal attention.

Evaluation: I really enjoyed reading and thinking about this paper, and think that questions about the extent that memory retrieval and attention overlap is of broad interest to the field. Additionally, I thought the analytic approach of training a pattern classifier on one task, and then asking how much brain activity during a separate task represented one mental state or another was novel and clever.

Despite my enthusiasm for many aspects of the paper, I'm not convinced that the results reported here support the broad conclusions drawn in the manuscript. Specifically, while it is intriguing and even surprising that EEG data during the attention task more closely resembles memory retrieval than encoding, I'm not convinced that we can conclude that internal attention explains the relationship. I also find it problematic that it was only possible to successfully classify retrieval vs encoding state in 35 of the 100 training observers. I've outlined these comments and a few others in more detail below:

I thank the Reviewer for their helpful comments and positive feedback on the analytic approach. I have revised the manuscript to address the Reviewer's concerns regarding the conclusions and the classification results. I respond to each concern below.

Primary comments:

- 1. As the manuscript highlights, there are many cognitive operations involved in both the attention and retrieval tasks. Unfortunately this feature of the task makes it difficult to conclude what cognitive process or processes underlies the observed similarities. I agree that both tasks rely on internal attention/working memory in some capacity. However, engaging in any computer based task also requires observers to direct their external attention to the computer display while they complete the task, even when the display is blank aside from fixation. Thus, a plausible alternative explanation is that observers are simply more likely to broaden their spotlight of external attention (or move the spotlight away from fixation) in the retrieval and attention tasks compared to the encoding task when they are more closely monitoring a shape at fixation.*

The Reviewer raises an interesting point and it is the case that participants may move the spotlight away from fixation during the retrieval and attention tasks. However, it is challenging to link spotlight broadening with the findings that (1) participants maintain cue direction and probe location during the delay and response intervals, (2) cue direction and probe location information is positively correlated with retrieval evidence, and (3) retrieval evidence predicts reaction times. First, a broader spatial spotlight should correspond to less evidence for either the right or left side of the screen, which is counter to the robust direction and location decoding observed in the present study. Second, if increased retrieval evidence reflects a widening spotlight then as retrieval evidence increases, the association with a precise spatial location (left or right) should diminish, thus the correlation between retrieval evidence and location information should be near zero or potentially negative. Finally, if increased retrieval evidence reflected a broader spotlight, the current results would indicate the broadest spotlight immediately prior to a behavioral decision. It is unclear how such a broad spotlight would facilitate responses when the *a priori* assumption would be that a narrow/focused spotlight, zoomed in on the probe identity/location,

would be optimal for behavioral performance.

The potential role of an attentional spotlight in the current study is directly related to recent theoretical work modeling retrieval as an internal attention process (Logan et al., 2021). Specifically, the authors developed a model with a spotlight of *internal* attention, analogous to the external attention spotlight which has been posited to support performance in the Eriksen flanker task. In addition to showing that a model with an internal attention spotlight captures episodic memory retrieval behaviors, the authors find that sharpening the attentional focus, i.e. narrowing the internal spotlight, reduces noise in the memory decision process and may facilitate memory performance by reducing attention to distracting information in memory. A prediction that follows from these findings is that during retrieve trials in the mnemonic state task the attentional spotlight would be narrow rather than broad. Therefore, I would interpret the current findings as reflecting that internal spotlight, whether to retrieved stimulus in the mnemonic state task or to maintained cue/probe information in the attention task. I have added text to the discussion (page 14) referencing this potential internal attention spotlight; the text is copied below.

Our interpretation is that participants direct their minds eye inwards at multiple points in a given trial in the attention task and that retrieval evidence tracks the extent to which participants have selected internal representations. This interpretation is in line with recent theoretical models in which an internal attentional spotlight supports memory retrieval by focusing attention to stored episodic representations²⁹.

- 2. Since it was only possible to distinguish between encoding and retrieval states for ~ 1/3 of observers, it doesn't seem like the encoding and retrieval task is a good candidate for measuring the relationship between retrieval and attention in the general population. That said, I might be missing something here. This concern could be alleviated by demonstrating that the same pattern of results is observed when all encoding and retrieval observers are included.*

To directly address this concern, I have re-run all of the mnemonic-state based analyses utilizing the full training (N=100) set of participants. The results from this re-analysis are qualitatively the same as is what is currently reported in the manuscript. I have reproduced the relevant statistics and figures below. All effects are in the same direction, but not all reach statistical significance. Given that the classifier has effectively been made 'worse' by the addition of data which do not consistently differentiate encoding and retrieval, this outcome is to be expected, as retrieval state estimates in the attention task will be noisier if the training set is noisier.

The proportion of participants that exceeded the 90th percentile threshold is consistent with prior work that I have conducted using the same mnemonic state task in an independent set of participants (Long & Kuhl, 2019). It is important to note that although a participant's within subject decoding accuracy may not exceed this threshold, it does not necessarily follow that they show no differentiation between encoding and retrieval states. Given that inclusion of all participants yields generally the same results suggests that most participants recruit similar mechanisms, but differ in the consistency with which they do so.

There are several potential explanations as to why only one third of participants exceed the threshold for inclusion in the cross-subject classifier. First, noisier data in general could interfere with classification and some participants may have noisier EEG signals than others. Second, participants could fail to follow instructions and not engage encoding or retrieval. As it is common practice to exclude participants with poor data quality and poor behavioral performance – and such participants were excluded

prior to the aggregation of the full training set of $N=100$ – these accounts are less likely, though of course not impossible. A more interesting alternative is that some participants may not engage memory states in the anticipated manner – i.e. solely engaging encoding when instructed to encode and solely engaging retrieval when instructed to retrieve. This may include the extent to which participants engage internally directed attention. As every trial has an externally presented stimulus that could be encoded and a categorically related stimulus that could be retrieved, participants could switch between encoding and retrieval during a single trial, preventing the classifier from both learning from that trial and correctly labeling it. Indeed, my lab has previously found that participants may automatically engage retrieval even when instructed to encode, before “correcting” and switching into a retrieval state (Smith et al., 2022). Investigations of memory state switching is outside of the scope of the present work, but may account for the variability in within-subject classification accuracy.

I chose to use the mnemonic state task to estimate memory states as it overcomes limitations common to classic memory paradigms. Memory encoding and memory retrieval are traditionally measured in different experimental phases with different response demands. These approaches limit the feasibility of developing a classifier as encoding vs. retrieval dissociations in these paradigms may be driven by aspects unrelated to memory processes, and instead may reflect perceptual demands (e.g. study phases always have external stimuli, test phases do not always have external stimuli) and/or response demands (e.g. stimuli can be studied without requiring participants to make a behavioral response, but behavioral responses are always required during the test phase). The mnemonic state task overcomes these limitations by closely matching perceptual and response demands across encode and retrieve trials. An object stimulus is presented on all trials and no behavioral responses are made on any trial.

Given these considerations, I have opted not to include the analyses and figures of the full mnemonic training set in the revised manuscript, but would be happy to include these data as supplementary results if the Reviewers feel that it would benefit the manuscript. I have added text to the methods (page 22) indicating that the proportion of above-threshold participants is in line with prior findings and that below-threshold participants may inconsistently recruit the same mechanisms as above-threshold participants.

The proportion of above threshold participants is in line with our prior work⁹. Participants who fall below threshold may not consistently engage encoding and retrieval states throughout the entire stimulus interval, although this does not necessarily mean that they do not engage the same neural mechanisms as the above-threshold participants.

Delay interval retrieval state evidence as a function of cue type, SOA, and time.

Effect	SOA = 200				SOA = 400				SOA = 800			
	df	F	p	η_p^2	df	F	p	η_p^2	df	F	p	η_p^2
Main effect of time	1,36	41.76	<0.0001	0.54	3,108	24.21	<0.0001	0.40	7,252	49.52	<0.0001	0.58
Main effect of cue	1,36	0.534	0.469	0.01	1,36	2.167	0.15	0.06	1,36	5.744	0.0219	0.14
Interaction of time × cue	1,36	0.008	0.927	0.0002	3,108	0.054	0.983	0.001	7,252	3.487	0.0014	0.09

Response interval retrieval state evidence as a function of cue type, SOA, and time.

Effect	df	Retrieval Evidence		
		F	p	η_p^2
Main effect of time	4,144	86.16	< 0.0001	0.71
Main effect of cue	2,72	1.681	0.193	0.04
Main effect of SOA	2,72	25.72	< 0.0001	0.42
Interaction of time × cue	8,288	1.576	0.131	0.04
Interaction of time × SOA	8,288	11.3	< 0.0001	0.24
Interaction of cue × SOA	4,144	3.481	0.0096	0.09
Interaction of cue × SOA × time	16,576	0.344	0.992	0.009

(A) Retrieval evidence varies over time

(B) SOA = 200 ms

(C) SOA = 400 ms

(D) SOA = 800 ms

(E) Cue direction and retrieval evidence are positively associated

Delay Interval Retrieval State Evidence. The mnemonic state classifier was trained utilizing the full set of $N=100$ participants who completed the mnemonic state task. Positive y-axis values indicate greater retrieval state evidence. The solid vertical line at time 0 to 100 ms indicates the onset of the cue. The vertical dashed lines indicate the onset of the probe, which varies as a function of stimulus onset asynchrony (SOA). **(A)** All trials are included; data have been averaged over cue type. Note that trial duration varies as a function of SOA, meaning that shorter SOA trials (200 and 400 ms) will end prior to the final time window shown. Across all SOAs, the trial initially begins with a decrease in retrieval that persists for approximately 500 ms, followed by an increase in retrieval that is maximal around the time point when the average response is made (indicated by the circles). **(B-D)** Each panel shows retrieval evidence separated by cue type (purple: cued, average of valid/invalid; grey: neutral) across the 100 ms cue and variable delay intervals. **(B)** There is no difference in retrieval state evidence between cued and neutral trials for the 200 ms SOA condition ($p = 0.469$). **(C)** There is no difference in retrieval state evidence between cued and neutral trials for the 400 ms SOA condition ($p = 0.15$). **(D)** Retrieval evidence is greater for cued compared to neutral trials for the 800 ms SOA condition ($p = 0.0219$). **(E)** We performed trial level Pearson correlations between cue direction evidence (left or right with neutral evidence as a baseline, see Methods) and retrieval state evidence. We find a significant correlation between cue direction evidence and retrieval state evidence for the 800 ms SOA condition. Error bars represent standard error of the mean. Box-and-whisker plots show median (center line), upper and lower quartiles (box limits), 1.5x interquartile range (whiskers) and outliers (diamonds). * $p < 0.05$

(A) Retrieval state evidence increases leading up to decision

(B) Probe location and retrieval evidence are positively correlated

(C) Retrieval and probe identity evidence predict reaction times

Response Interval Retrieval State Evidence. The mnemonic state classifier was trained utilizing the full set of $N=100$ participants who completed the mnemonic state task. **(A)** Each panel shows probe-locked retrieval state evidence; positive values indicate greater retrieval state evidence. The dashed vertical line at time 0 to 100 ms indicates the onset of the probe (cross target or plus lure). The left panel shows retrieval state evidence separated by cue type (invalid, red; neutral, grey; valid, blue). The middle panel shows retrieval state evidence separated by SOA (200, 400, 800ms). The right panel shows retrieval state evidence separately for hits (cross targets to which participants responded; dark green) and correct rejections (plus lures to which participants withheld a response; light green). **(B)** We performed trial level Pearson correlations between probe identity (cross, plus) evidence (left panel) or probe location (left, right) evidence (right panel) and retrieval state evidence across the response interval. We find a significant positive correlation between probe location and retrieval evidence during the 200-500ms of the response interval. **(C)** We performed multiple linear regression in which we used retrieval evidence and probe identity evidence during the 300-400ms time window to predict reaction times (RTs). We only included trials with RTs > 400 ms. We find significant negative betas for both regressors meaning that more retrieval evidence and more probe identity evidence predict faster RTs. Error bars represent standard error of the mean. Box-and-whisker plots show median (center line), upper and lower quartiles (box limits), 1.5x interquartile range (whiskers) and outliers (diamonds). * $p < 0.05$; ** $p < 0.01$; *** $p < 0.001$.

3. *The current results tell us that the brain activity during the attention task is more like retrieval trials than encoding trials, but doesn't tell us how similar activity is between the attention and retrieval task. One way to test this, is to conduct a cross subject decoding analysis that includes all three tasks. An observation of equal retrieval and attention classifier evidence for retrieval and attention trials, would provide strong evidence for the conclusion that both tasks rely on the same cognitive process.*

I thank all three Reviewers for raising this very important point and I agree that the strictest test of the overlap between retrieval and internal attention is through cross-task classification. The current study is not ideally suited for cross-task decoding of encoding vs. retrieval vs. attention given that the mnemonic data are within subject (the same participants performed encoding and retrieval) whereas the cross-study application is between subject. Additionally, to the extent that such a task exists, the attention task in the current manuscript is not a “pure” internal attention task. Instead, attentional demands are expected to change between external (to cue and probe) and internal (to maintained information). The attention task conditions – cue type and delay – also impact the degree and temporal dynamics of attention allocation, meaning that attention is likely to vary across time and trials in the attention task. Together, as laid out by Reviewer 3, the strongest approach would be to utilize multiple specific retrieval and internal attention tasks and demonstrate successful cross-task classification which is best suited to future work. Therefore, I have toned down the interpretation of the current data by making edits throughout the text (including the title), and specifically address this limitation and potential future direction in the discussion (page 16), text copied below.

A critical next step will be to perform direct cross-task classification on a set of a retrieval and internal attention tasks in order to determine the extent to which the two tasks overlap. Robust cross-task classification would provide strong evidence that the retrieval state and internal attention are one and the same, whereas the current findings leave open the possibility that specific processes beyond internal attention are unique to the retrieval state.

4. *The reported analyses always train and test on the same time points (i.e., train at 100 ms in the encoding/retrieval task and test at 100 ms in the attention task). Since the tasks themselves are so different, matched time points may not provide the best decoding. A straightforward way to test this would be to include a temporal generalizability analysis where each time point of the training task is used to classify each time point of the testing task (see MacLean et al., 2019 figure 3 for a nice example).*

The Reviewer is correct that there is no *a priori* reason to expect processes to unfold at the same time points in the two different tasks and that matched time points may not provide the best decoding. However, the analyses reported in the manuscript do not train and test on the same time points. Instead, the classifier was trained to distinguish encode vs. retrieve states based on spectral power averaged across the 2000 ms stimulus interval. The rationale for this approach is that this time interval encompasses all possible time points at which memory encoding and retrieval may be dissociated without making any assumptions about which time interval(s) may yield the strongest encoding vs. retrieval dissociations, which is/are likely to vary across participants. Importantly, cross-subject decoding is successful for the 2000 ms stimulus interval. If cross-subject classification failed (i.e. if cross-subject classification accuracy did not differ from chance), the classifier could not be applied to the attention task data. In order to perform the temporal generalizability analysis suggested by the Reviewer, it would be necessary to validate that each 100 ms time interval yields robust cross-subject memory state decoding. To the extent that memory state engagement is variable over time, cross-subject decoding at such fine time scales will be infeasible. The variability in timing of memory state engagement is a potentially interesting question, but is outside of the scope of the present manuscript. I have clar-

ified the rationale for the training interval selection in the methods (page 21) and I have added text to the discussion regarding individual differences in temporal dynamics of the retrieval state (page 17). I have copied the relevant text below.

We utilized the full stimulus interval as we have previously found robust within-subject decoding during this interval⁹ and did not have a priori predictions regarding which time interval(s) would have the strongest memory state dissociations. (Page 21)

Furthermore, the present work raises questions regarding the temporal dynamics of memory brain states, including how individuals transition into and out of these brain states and the consequences for behavior of switching vs. staying in a particular brain state. (Page 17)

5. *Interpreting the cross-task decoding results requires a clear understanding of the encoding vs retrieval decoding performance, so it would be helpful to include a plot of the time-resolved encoding vs retrieval classification results.*

References:

MacLean MH, Bullock T, Giesbrecht B (2019) Dual Process Coding of Recalled Locations in Human Oscillatory Brain Activity. The Journal of Neuroscience 39:6737.

I agree with the Reviewer and have added a histogram showing the mnemonic state classification results (Figure 2A; page 32). I have reproduced this figure panel below.

Cross-subject mnemonic state decoding. We trained an L2-logistic regression classifier to discriminate encoding versus retrieval memory states across 35 participants. The classifier was trained and tested on spectral signals from 63 electrodes and 46 frequencies, averaged across the 2000 ms stimulus interval. Mean classification accuracy across all subjects (solid vertical line) is shown along with a histogram of classification accuracies for individual subjects (gray bars) and mean classification accuracy for permuted data across all subjects (dashed vertical line). Mean classification accuracy was 60%, which differed significantly from chance ($p < 0.0001$).

Reviewer #3 (Remarks to the Author):

Review of “Reconceptualizing the retrieval state as an internal attention state”

1. *This paper tests the hypothesis that “retrieval state” (i.e., whether you use a stimulus as a prompt to retrieve a related memory, vs. encoding the stimulus as a new memory) can be conceptualized as internally-directed attention. The author’s strategy for testing this hypothesis is to take a “retrieval state” EEG classifier (trained on an already-collected dataset), apply it to Posner-like cued spatial attention task, and then assess whether the output of the classifier aligns with how we think internally-directed attention would behave in that task. Essentially, the approach here is to build an extremely detailed “circumstantial” case for the equivalence between the processes – the paper analyzes how retrieval state evidence varies as a function of a wide range of factors (e.g., stimulus onset asynchrony, cue type) and how it relates to other kinds of classifier evidence (e.g., for cue direction, probe location, and probe identity) across all of the different time epochs of the trial, and then argues how each of these findings is what you would expect if retrieval state evidence were tracking internally-directed attention. The case that the author builds is very thorough and the total weight of evidence in support of the author’s argument is strong; while it’s not clear if the two processes are exactly the same (see paragraph below), this paper shows that, at the very least, they are quite closely related. As such, I think it’s an important contribution. I also am a big fan of the approach of taking a pre-trained classifier from another task and applying it to a new task – as illustrated by the present study, this process will allow our field to get an increasingly refined sense of what the classifier measures. Over time, I can imagine the field accumulating a “library” of well-validated classifiers that each tap into distinct processes.*

I thank the Reviewer for their positive feedback on the manuscript and their helpful comment regarding the interpretation of the current finding. I provide my response below.

My only substantive comment relates to the challenges inherent in establishing the “identity” of retrieval state and internally-directed attention. As noted above, the two processes clearly (from the results shown here) have many elements in common, but the approach taken in this paper can not rule out the possibility that there are some subprocesses related to retrieval state that are not related to internally-directed attention, or vice-versa. In an ideal world, one could have multiple tasks tracking retrieval state (say, A and B) and multiple tasks tracking internally-directed attention (say, C and D) and then show that all possible pairs of tasks cross-generalize equally well (e.g., training on a retrieval state task generalizes to internally-directed attention task C as well as training on internally-directed attention task D generalizes to internally-directed attention task C; and it would also be useful to show that training on an internally-generated attention task generalizes to retrieval state task A as well as training on retrieval state task B generalizes to retrieval state task A). Having said this, I don’t think it’s reasonable to expect the author to do something like this in the scope of this paper; in my view, the contribution is strong enough as it stands. Consequently, I would be satisfied with the author adding a few caveats about the kinds of inferences about process identity that are licensed by the design they used.

Sincerely,

Ken Norman (I sign all of my reviews)

I thank all three Reviewers for raising this very important point and I agree that the strictest test of the overlap between retrieval and internal attention is through cross-task classification. The current study is not ideally suited for cross-task decoding of encoding vs. retrieval vs. attention given that the mnemonic data are within subject (the same participants performed encoding and retrieval) whereas

the cross-study application is between subject. As the Reviewer describes, the strongest approach would be to utilize multiple specific retrieval and internal attention tasks and demonstrate successful cross-task classification which is best suited to future work. Therefore, I have toned down the interpretation of the current data by making edits throughout the text (including the title), and specifically address this limitation and potential future direction in the discussion (page 16), text copied below.

A critical next step will be to perform direct cross-task classification on a set of a retrieval and internal attention tasks in order to determine the extent to which the two tasks overlap. Robust cross-task classification would provide strong evidence that the retrieval state and internal attention are one and the same, whereas the current findings leave open the possibility that specific processes beyond internal attention are unique to the retrieval state.

Reviewers' Comments:

Reviewer #1:

Remarks to the Author:

The author has revised the manuscript in a way that eliminated my concerns. I have one minor note regarding my comment regarding the classifier picking up on spatial selective attention: My suggestion was not that the classifier picks up a particular location. Instead, I argue that it can pick up internal spatial attention to any location. In other words, the common aspect of retrieval and Posner's task may include selectively attending to a spatial location. However, the author already specified that the internal attention component that is described in the paper is regarding selective attention. Such selective attention being spatial is speculative, therefore does not need to be included in the paper.

Reviewer #2:

Remarks to the Author:

The author did an excellent job addressing the concerns I raised in my initial review, and I think the current manuscript will make a great addition to the literature. At this point I only have a few suggestions for the author that they should feel free to take or leave.

1. Seeing that the main findings of the manuscript mostly hold up when all observers are included in the training set moved my impression of the manuscript from skeptical that the results tell us anything about how memory and attention are related in the general population, to reasonably confident that this effect will hold up in future work. Having the full training sample results that were included in the response to reviewers available in the supplementary materials would likely help convince readers who are similarly skeptical about the generalizability of findings based on a small subset of the training task participants.

2. As I was reading, I often found myself needing to refer back to the mnemonic state task paper so I could think through the relationship between the tasks. Adding a task figure for the mnemonic state task (and perhaps a classifier training/testing procedure visualization) could help reduce the working memory load for future readers.

Sincerely,

David Sutterer

Reviewer #3:

Remarks to the Author:

I am satisfied with the revisions -- I think this is a very nice contribution to the literature.

Sincerely,

Ken Norman

May 6, 2023

Dear Editors:

Thank you for the opportunity to revise the manuscript, "The intersection of the retrieval state and internal attention." I have made revisions to the manuscript in response to these comments and provide an item-by-item summary of the revisions to the Reviewers' concerns (italicized and in blue; responses in black) in the pages below.

Reviewer #1 (Remarks to the Author):

The author has revised the manuscript in a way that eliminated my concerns. I have one minor note regarding my comment regarding the classifier picking up on spatial selective attention: My suggestion was not that the classifier picks up a particular location. Instead, I argue that it can pick up internal spatial attention to any location. In other words, the common aspect of retrieval and Posner's task may include selectively attending to a spatial location. However, the author already specified that the internal attention component that is described in the paper is regarding selective attention. Such selective attention being spatial is speculative, therefore does not need to be included in the paper.

I thank the Reviewer for clarifying this point.

Reviewer #2 (Remarks to the Author):

The author did an excellent job addressing the concerns I raised in my initial review, and I think the current manuscript will make a great addition to the literature. At this point I only have a few suggestions for the author that they should feel free to take or leave.

I thank the Reviewer for their helpful comments and I respond to each concern below.

Primary comments:

- 1. Seeing that the main findings of the manuscript mostly hold up when all observers are included in the training set moved my impression of the manuscript from skeptical that the results tell us anything about how memory and attention are related in the general population, to reasonably confident that this effect will hold up in future work. Having the full training sample results that were included in the response to reviewers available in the supplementary materials would likely help convince readers who are similarly skeptical about the generalizability of findings based on a small subset of the training task participants.*

I have included the full training sample results in the supplementary materials.

- 2. As I was reading, I often found myself needing to refer back to the mnemonic state task paper so I could think through the relationship between the tasks. Adding a task figure for the mnemonic state task (and perhaps a classifier training/testing procedure visualization) could help reduce the working memory load for future readers.*

Sincerely,

David Sutterer

I have added a figure (Figure 2) showing the mnemonic state task and a schematic outlining the cross-study classification approach.

Reviewer #3 (Remarks to the Author):

I am satisfied with the revisions – I think this is a very nice contribution to the literature.

*Sincerely,
Ken Norman*

I thank the Reviewer for their feedback.